# Addition of a fast GC to SIFT-MS for analysis of individual monoterpenes in mixtures

Michal Lacko[1,2], Nijing Wang[3], Kristýna Sovová[1], Pavel Pásztor[1], Patrik Španěl[1]

[1]The Czech Academy of Science, J. Heyrovský Institute of Physical Chemistry, Dolejškova 2155/3, 182 23 Prague, Czech Republic
[2]Faculty of Mathematics and Physics, Charles University in Prague, Ke Karlovu 3, 121 16 Prague, Czech Republic
[3]Air Chemistry Department, Max-Planck-Institut für Chemie, Hahn-Meitner-Weg 1, 55128 Mainz, Germany

*Correspondence to*: Michal Lacko (michal.lacko@jh-inst.cas.cz)

**Abstract.** Soft chemical ionization mass spectrometry (SCI-MS) techniques can be used to accurately quantify volatile organic compounds (VOCs) in air in real time; however, differentiation of isomers still represents a challenge. A suitable pre-separation technique is thus needed, ideally capable of analyses in a few tens of seconds. To this end, a bespoke fast GC with an electrically heated 5 m long metallic capillary column was coupled to selected ion flow tube mass spectrometry (SIFT-MS). To assess the performance of this combination, a case study of monoterpene isomer ($C_{10}H_{16}$) analyses was carried out. The monoterpenes were quantified by SIFT-MS using $H_3O^+$ reagent ions (analyte ions $C_{10}H_{17}^+$, *m/z* 137, and $C_6H_9^+$, *m/z* 81) and $NO^+$ reagent ions (analyte ions $C_{10}H_{16}^+$, *m/z* 136, and $C_7H_9^+$, *m/z* 93). The combinations of the fragment ion relative intensities obtained using $H_3O^+$ and $NO^+$ were shown to be characteristic of the individual monoterpenes. Two non-polar GC columns (Restek Inc.) were tested: the advantage of MXT-1 was shorter retention whilst the advantage of MXT-Volatiles was better separation. Thus, it is possible to identify components of a monoterpene mixture in less than 45 s by the MXT-1 column and to separate them in less than 180 s by the MXT-Volatiles column. Quality of separation and sensitivity of present technique (LOD ~16 ppbv) was found to be inferior compared to commercially available fast-GC solutions coupled with proton transfer reaction mass spectrometry (PTR-MS, LOD ~1 ppbv) due to the limited sample flow through the column. However, using combinations of two reagent ions improved identification of monoterpenes not well resolved by the chromatograms. As an illustrative example, the headspace of needle samples of three conifer species was analysed by both reagent ions and with both columns showing that mainly α-pinene, β-pinene and 3-carene were present. The system can thus be used for direct rapid monitoring of monoterpenes above 20 ppbv, such as applications in laboratory studies of monoterpene standards and leaf headspace analysis. Limitation of the sensitivity due to the total sample flow can be improved using a multicolumn pre-separation.

## 1 Introduction

Monoterpenes, mostly emitted from plants, are very important biogenic volatile organic compounds (BVOCs) in the atmosphere. Due to their high reactivity with atmospheric oxidants such hydroxyl radicals (OH•), oxidation of monoterpenes

can lead to tropospheric ozone ($O_3$) accumulation as well as to secondary organic aerosol formation, which can affect human health and contribute to global climate change (Chameides et al. (1992); Fehsenfeld et al. (1992); Kulmala et al. (2004)). Although all monoterpenes comprise two isoprene units and all have the same molecular formula, $C_{10}H_{16}$, their lifetime (inverse to reactivity) for reaction with $OH^{\bullet}$ and $O_3$ widely varies from minutes to days (Atkinson and Arey, 2003) (See Table 1). The values of the total OH reactivity, which is dominated by BVOCs measured in rainforests, have been found to be higher than expected, which could be attributed to undetected monoterpenes or sesquiterpenes (Nolscher et al., 2016). Therefore, it is important to identify and individually quantify these BVOCs at their ambient trace levels.

**Table 1. Monoterpenes included in the present study listed together with their atmospheric lifetime.**

| Compound | Lifetime for reaction with a $OH^b$ $O_3{}^c$ | Chemical lifetime[d] | | Rate constant of $O_3$ reaction[e] | Rate constant of OH reaction[f] |
|---|---|---|---|---|---|
| | | *Day* | *Night* | | |
| α-pinene | 2.6 hrs 4.6 hrs | 2-3 hrs | 5-30 min | 8.7 | 5.45 ± 0.32 |
| β-pinene | 1.8 hrs 1.1 day | 2-3 hrs | 5-30 min | 1.5 | 7.95 ± 0.52 |
| camphene | 2.6 hrs 18 day | nd | nd | 9.0[g] | 5.33[g] |
| myrcene | 39 min 50 min | 40-80 min | 5-20 min | 49 | 21.3 ± 1.6 |
| 3-carene | 1.6 hrs 11 hrs | nd | nd | 3.8 | 8.70 ± 0.43 |
| (R)-limonene | 49 min 2.0 hrs | 40-80 min | 5-20 min | 21 | 16.9 ± 0.5 |
| α-terpinene | 23 min 1 min | < 5 min | < 2 min | 870 | 36.0 ± 4.0 |
| γ-terpinene | 47 min 2.8 hrs | nd | nd | 14 | 17.6 ± 1.8 |

[a] taken from Atkinson (Atkinson and Arey, 2003) unless noted otherwise.
[b] Assumed OH radical concentration: $2.0 \times 10^6$ molecule $cm^{-3}$, 12-h daytime average.
[c] Assumed $O_3$ concentration: $7 \times 10^{11}$ molecule $cm^{-3}$, 24-h average.
[d] Lifetimes are estimated in relation to $[NO_3] = 10$ pptv, $[O_3] = 20$ ppb for night; and $[OH] = 10^6$ molecules per $cm^3$, $[O_3] = 20$ ppb for day light conditions. (Kesselmeier and Staudt, 1999) (unless noted otherwise)
[e] Rate constants (in units of $10^{-17}$ $cm^3$ $molecule^{-1}$ $s^{-1}$) for the gas-phase reactions of $O_3$ with a monoterpenes have been determined at $296 \pm 2$ K and 740 Torr total pressure of air or $O_2$ using a combination of absolute and relative rate techniques. (Atkinson et al., 1990) (unless noted otherwise)
[f] Rate constants (in units of $10^{-11}$ $cm^3$ $molecule^{-1}$ $sec^{-1}$) for the gas-phase reactions of the OH radical with monoterpenes have been determined in one atmosphere of air at $294 \pm 1$ K. (Atkinson et al., 1986) (unless noted otherwise)
[g] Rate constants of k(OH + isoprene) = $1.01 \times 10^{-10}$ $cm^3$ $molecule^{-1}$ $s^{-1}$. $O_3$ reaction rate constants determined in $10^{-19}$ $cm^3$ $molecule^{-1}$ $s^{-1}$ units. OH radical reaction rate constants determined in $10^{-11}$ $cm^3$ $molecule^{-1}$ $s^{-1}$ units. (Atkinson et al., 1990)
nd – no data.

Standard analytical methods used to identify and quantify volatile organic compounds (VOCs) in air, such as thermal desorption gas chromatography mass spectrometry (TD-GC-MS), are often time consuming and cannot be used to investigate temporal changes in chemically evolving systems. In contrast, soft chemical ionization mass spectrometry (SCI-MS) techniques, such as selected ion flow tube mass spectrometry (SIFT-MS) (Smith and Španěl, 2011a; Španěl et al., 2006) and proton transfer reaction mass spectrometry (PTR-MS) (Lindinger et al., 1998; Ellis and Mayhew, 2013; Smith and Španěl, 2011b) represent well-established real time tools to analyse a wide variety of VOCs in ambient air (Amelynck et al., 2013; de Gouw and Warneke, 2007; Malásková et al., 2019; Rinne et al., 2005; Schoon et al., 2003) and in headspace of biological samples (Shestivska et al., 2015; Shestivska et al., 2011; Shestivska et al., 2012). The advantage of SIFT-MS and PTR-MS lies in the possibility of online, real-time analysis obviating sample collection and pre-concentration of VOCs. In these techniques, defined reagent ions (usually $H_3O^+$, $NO^+$ or $O_2^{+\bullet}$) interact with trace VOCs present in gas samples introduced into a flow tube or a flow/drift tube. The chemical ionisation reactions that produced analyte ions are variously proton transfer, adduct ion formation, charge transfer and hydride ion transfer, principally depending on the type of reagent ions used. This ion chemistry has been thoroughly reviewed in a number of publications (Smith and Španěl, 2005). These ion-molecule reactions are not greatly exothermic thus few product (analyte) ions result in each reaction, often just one or two, that can readily be identified. However, chemically similar molecules with the same atomic composition (structural isomers) usually produce identical analyte ions with similar branching ratios and therefore the neutral analyte molecules cannot be easily differentiated using SCI-MS alone (Smith et al., 2012). As a result, standard SCI-MS techniques such as SIFT-MS and PTR-MS are limited to reporting concentrations of the sum of monoterpenes presented in the sample, and the composition of the monoterpenes present cannot be determined. However, the reactions of the isomeric molecules may have different rate coefficients with the different reagent ions and lead to product ions at recognisably different branching ratios depending on their molecular geometry (Jordan et al., 2009; Pysanenko et al., 2009; Španěl and Smith, 1998; Wang et al., 2003). So the concurrent use of the available reagent ions in SIFT-MS analysis can sometimes be used to analyse and identify particular isomers.

Quantitative measurement of monoterpenes is often problematic due to problems of stability of monoterpene mixtures in certified gas standards (Rhoderick and Lin, 2013). Therefore, fresh individual monoterpene standards or monoterpene mixtures can be prepared from liquid standards. To determine an accurate instrument sensitivity to individual monoterpenes, the relative abundance of monoterpene isomers must be known (de Gouw et al., 2003).

Gas chromatography mass spectrometry (GC-MS) coupled with pre-concentration techniques has been developed to successfully identify and quantify different atmospheric monoterpenes (Janson, 1993; Räisänen et al., 2009; Song et al., 2015). However, the requirements of pre-concentration and long cycle time (more than 1h) are obviously unsuitable for real-time measurements.

A promising approach to the near real time analysis of isomeric molecules is to combine both SCI-MS and fast GC methods. Pre-separation provided by fast GC requires short columns with thin active layers, fast temperature ramps, fast injection systems and time resolutions below 5 min (Matisová and Dömötörová, 2003). Materic et al. (Materić et al., 2015) established

a system using PTR-MS coupled with a fast GC to detect individual monoterpenes and achieved the separation of six most common monoterpenes at a limit of detection down to 1 ppbv. Pallozzi et al. then compared a fastCG-PTR-ToF-MS system with traditional GC-MS methods, discussing the limitations of the fast GC peak separation on some BVOCs emitted from plants, including monoterpenes (Pallozzi et al., 2016). The authors then recommended applying longer columns operating with

5    fast temperature gradient such as 25 °C min[-1]. SIFT-MS is also widely used in VOCs analyses (Allardyce et al., 2006; Smith and Španěl, 2011b, 2005b), which has well-defined analytical reaction conditions and the $H_3O^+$, $NO^+$ and $O_2^{+\bullet}$ reagent ions can be switched rapidly to analyse time-varying trace gas concentrations in air samples. In the present article, we report experimental developments aimed at selectively analysing individual monoterpenes in mixtures in air using a bespoke fast GC/SIFT-MS combination with $H_3O^+$ and $NO^+$ reagent ions. This involved the analysis of both prepared laboratory

10   monoterpene/air mixtures and the headspace of the foliage of different pine trees.

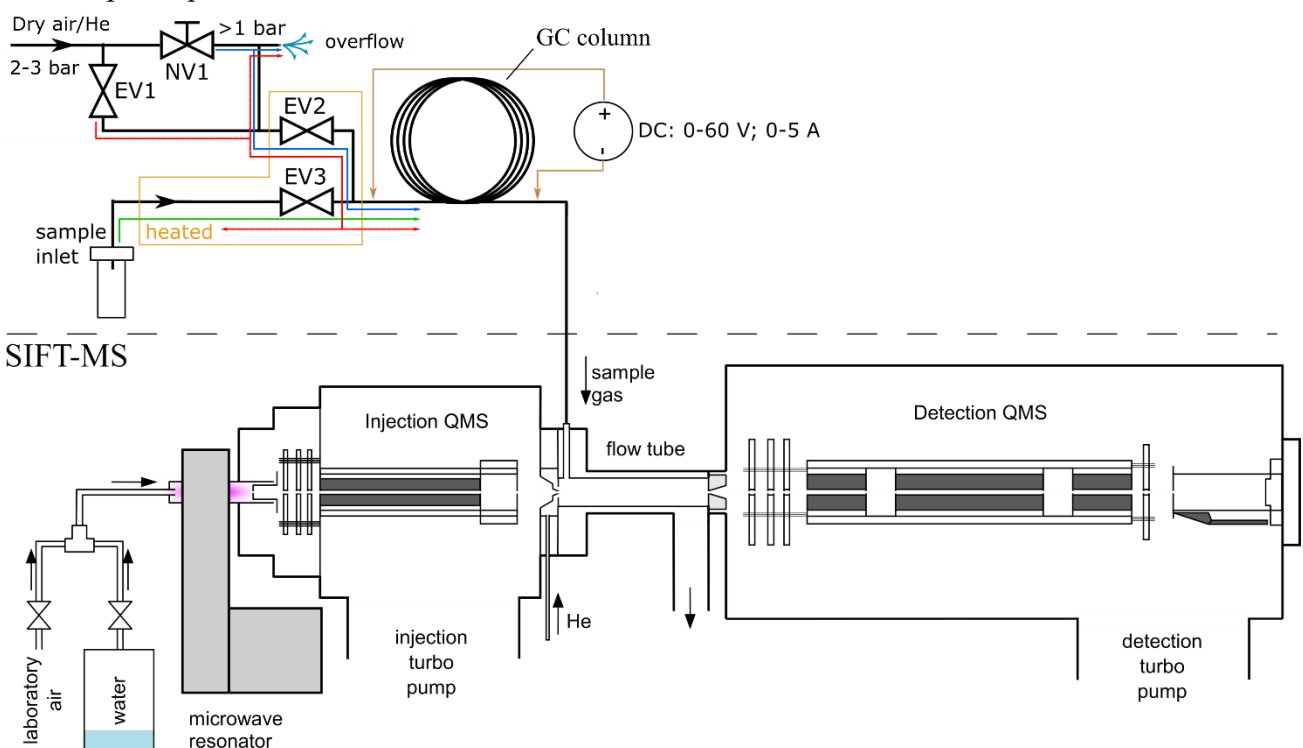

**Figure 1: Schematic visualization of the fast GC/SIFT-MS experiment. Coloured lines in the inlet part of the fast CG represent gas flow through the system of the valves EV1-3. The blue line traces the "normal mode" regime, the green line represents the "sampling mode" and the red line represents the "cleaning mode".**

## 2 Construction of a fast GC device for pre-separation

The experimental setup of the bespoke fast GC setup constructed as an addition to SIFT-MS is shown in Fig. 1. The routing of the sample and the carrier gases was controlled by solenoid valves (Parker VSONC-2S25-VD-F, < 30ms response), labelled in Fig. 1 as EV1, EV2 and EV3. The needle valve NV1 was used in combination with an overflow relieve tube to fine-adjust the flow rate of the carrier gas (20-50 sccm from a gas cylinder, regulator set to about 2 bar) so that the air pressure at the column entrance is held just above ambient. The region of the sampling input line, EV2, EV3 and their connection with the column are permanently heated to ~60 °C to prevent adsorption of sample gas/vapour and to reduce memory effects.

Three modes of gas flow are possible as illustrated in Fig. 1:

- The **"normal mode"**: EV2 is open and both EV1 and EV3 are closed. Carrier gas flows through NV1, partly vented via the overflow relief but mostly into the column. The pressure at the column entrance is just above the ambient atmosphere and a constant flow rate of clean carrier gas (synthetic air or helium) is thus achieved.

- The **"sampling mode"**: EV1 and EV2 are closed and EV3 is open. Sample air is introduced into the column in a short time (1 to 12 s) after which the "normal mode" is resumed.

- The **"cleaning mode":** All valves are open and the carrier gas taken directly from the cylinder regulator is introduced into the column (higher than normal flow) and purges the sample line via EV3. The overflow relieve flow rate is not sufficient to diminish the pressure.

The modes can be switched either manually or controlled from the SIFT-MS software.

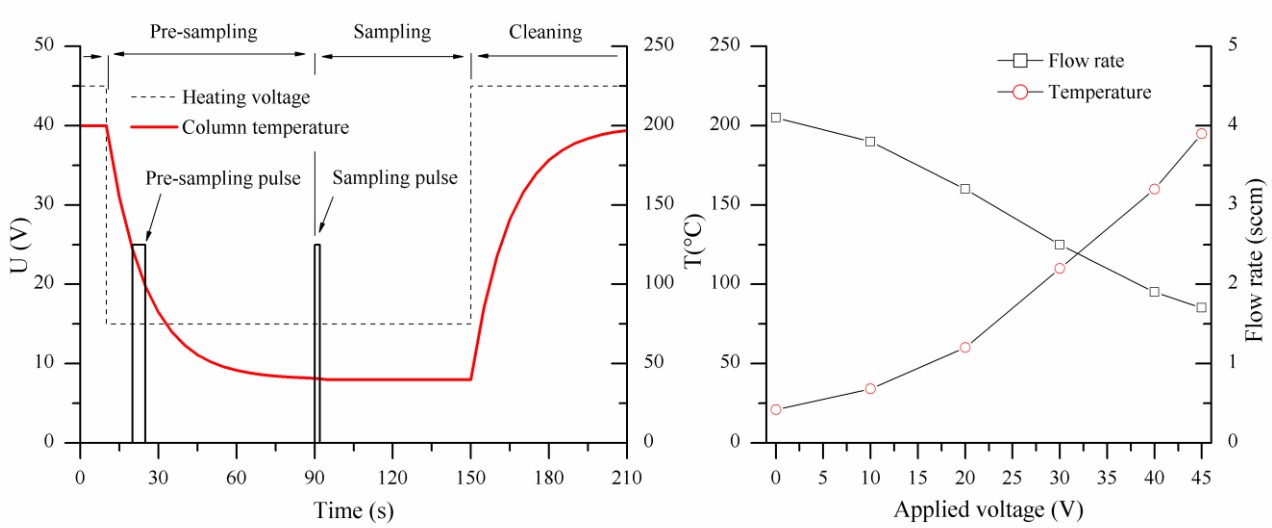

**Figure 2: Left: the applied heating voltage (dashed) and the temperature profile of the column (red) during the fast GC cycle. The pulses indicate the opening of the valve EV3 during the pre-sampling and the sampling periods. Right: The increase of the column temperature and the related decrease of the carrier gas flow rate with the heating voltage.**

The operation sequence for air sample analysis is as follows: The column is first heated up to 200 °C in the "cleaning mode" for three minutes prior to commencing the "normal mode" with an appropriate heating voltage setting (e.g. 15 V as shown in Fig. 2). Whilst the column cools down, a pre-sampling interval (8-10 s "sampling mode", see Fig. 2) is applied in order to refill the "dead volume" comprising the EV3 valve and the sampling inlet by air at its entrance. After the column reaches working temperature and a steady flow of clean carrier gas is established, the sample for actual analysis is introduced by enabling the "sampling mode" for a selected time period.

In the experiments, two different GC columns were tested. First, a 5 m long nonpolar general-purpose chromatography metallic column MXT-1 (0.28 mm × 0.1 μm active phase, Restek Inc.) using dry air as the carrier gas. The column was chosen according to the previous PTR-MS fastGC analyses (Romano et al., 2014). Additionally, a second, application-specific column for volatile organic pollutants, MXT-Volatiles (0.28 mm × 1.25 μm active phase, Restek Inc.), was used with helium carrier gas. In order to facilitate direct resistive heating, the coil-shaped stainless steel columns (resistivity ~4.2 $\Omega$ m$^{-1}$) were electrically isolated and connected to a regulated 60 V, 5 A DC power supply. Appearance of cold spots was suppressed by ensuring that the electrical current runs through the entire length of the columns. The temperatures of the columns were monitored by a K-type probe connected to their centres (see the right part of Fig. 2 for the temperature variation with applied voltage). It is interesting to note that the flow of sampled air, established by the pressure difference between ambient atmosphere and the low pressure of the SIFT-MS flow tube, changes with the column temperature due to the variation of the dynamic viscosity of the air (see Fig. 2). This effect can be estimated by direct measurement of the column flow rate and has to be accounted for in the quantification calculation (see Eq. 8).

In the initial tests with the first generic MXT-1 column, the "sampling mode" duration was fixed at 1.8 s due to SIFT-MS software limitations. For the later tests with the second MXT-Volatiles column, the SIFT-MS operational software was upgraded to provide an arbitrary timing of the "sampling mode" duration, where 6 or 12 s sampling intervals were used. Sampling was repeated several times to improve signal quality. The GC separation then takes place over typically 60 – 300 s whilst the eluent is continuously analysed by SIFT-MS. It is possible to apply a heating ramp during this period.

Several heating ramp profiles were tested (see data for MXT-1 column in Fig. S1 in the Supplement); however, due to the short GC column and relatively long injection time, the monoterpene chromatogram peaks coalesced when the column temperature exceeded 60 °C and it was found that optimal chromatograms were obtained isothermally at 40 °C (15 V heating voltage). Effects of the heating voltage on the retention time and the chromatogram profile are illustrated in Fig. S4 in the Supplement (data for MXT-Volatiles column).

## 3 SIFT-MS analyses of the eluent

In the present study, a *Profile 3* SIFT-MS instrument (Instrument Science, Crewe, UK) was used (Smith et al., 1999). Reagent ions are formed in a microwave discharge through a mixture of water vapour and atmospheric air (see Fig. 1). A mixture of ions is extracted from the discharge and focused into a quadrupole mass filter where they can be analysed according to their

mass-to-charge ratio, $m/z$. Thus, the reagent ions $H_3O^+$, $NO^+$ or $O_2^{+\bullet}$ can be selected ($O_2^{+\bullet}$ was not used in the present experiment) and separately injected into flowing helium carrier gas (pressure $p = 1.4$ mbar, temperature $T = 24$ °C). Any internal energy possessed by the reagent ions is rapidly quenched in collisions with helium atoms leaving a thermalized ion swarm that is convected down the flow tube. Sample gas is introduced into the helium/thermalized swarm at a known flow

rate that (in the present experiments) changes with the GC column temperature. The reagent ions react with the VOC molecules in the sample gas during a time period defined by the known flow speed of the ion swarm and the length of the flow tube. At the end of the flow tube, the ionic products (analyte ions) generated by the analytical ion-molecule reactions are sampled by a pinhole orifice into the analytical quadrupole mass spectrometer. The count rates of the reagent and analyte ions are obtained using a single channel electron multiplier. Thus, full scan (FS) spectra can be obtained over a chosen $m/z$ range to identify the

analyte ions or rapidly switched between selected $m/z$ values using the multiple-ion monitoring mode (MIM) (Španěl and Smith, 2013; Smith and Španěl, 2011a). For the present monoterpene study, the FS mode was used for SIFT-MS analyses, whilst the MIM mode was used for fast GC/SIFT-MS setup. The typical count rate of the reagent ions was one million per second, cps, while those for the analyte ions were usually below 1 cps. Switching between the $H_3O^+$ and $NO^+$ reagent ions required a few milliseconds, depending mainly on the velocity of the carrier gas (12 000 cm s$^{-1}$) and the length of the flow tube

(5 cm). Therefore, the only limiting factor is the software sampling frequency, which depends on the number of monitored ions, but is usually below one second.

### 3.1 Reactions of the $H_3O^+$ and $NO^+$ reagent ions with monoterpenes

In the present study, SIFT-MS analyses of monoterpenes were carried out using the previously investigated reactions of monoterpenes with $H_3O^+$ and $NO^{+\cdot}$ ions (Schoon et al., 2003; Wang et al., 2003). The $H_3O^+$ reactions are known to proceed via

proton transfer forming $C_{10}H_{17}^+$ ($m/z$ 137) ions that partially fragments to $C_6H_9^+$ ($m/z$ 81) by the elimination of a $C_4H_8$ moiety from the nascent $(C_{10}H_{17})^*$ excited ion:

$$H_3O^+ + C_{10}H_{16} \rightarrow C_{10}H_{17}^+ + H_2O \tag{1a}$$
$$\rightarrow C_6H_9^+ + C_4H_8 + H_2O \tag{1b}$$

The known values of the proton affinities (PA) of α-pinene, camphene (both 878 kJ mol$^{-1}$) (Solouki and Szulejko, 2007), and

(R)-limonene (875 kJ mol$^{-1}$) (Fernandez et al., 1998) are well above the PA of water (691 kJ mol$^{-1}$) (NIST). The excess energy following proton transfer (almost 2 eV) allows the observed dissociation to occur.

$NO^+$ reacts with monoterpenes by charge transfer forming the parent cation $C_{10}H_{16}^{+\bullet}$ ($m/z$ 136) and a number of fragment ions, including $C_7H_9^+$:

$$NO^+ + C_{10}H_{16} \rightarrow C_{10}H_{16}^+ + NO \tag{2a}$$

$$\rightarrow C_7H_9^+ + NOC_3H_7 \tag{2b}$$

The exothermicity of charge transfer (2a) is represented by the difference between the ionization energies of the neutral NO (9.26 eV) and that for the particular monoterpene (ranging from 8.07 eV for α-pinene to 8.4 eV for (R)-limonene) (Garcia et

al., 2003; NIST). Other fragment ions, including $C_7H_8^+$, $C_7H_{10}^+$, $C_9H_{13}^+$ and $C_{10}H_{15}^+$, are also seen and the branching ratios between the channels (2a) to (2b) and other fragments depend on the isomeric structure of the monoterpene (Schoon et al., 2003; Wang et al., 2003). The branching ratios are given in Table S1 in the Supplement. Based on this known ion chemistry, for the present study, the monoterpenes analysis was accomplished using both the $H_3O^+$ reagent ions (recording the $C_{10}H_{17}^+$

($m/z$ 137) and $C_6H_9^+$ ($m/z$ 81) analyte ions) and $NO^+$ reagent ion (recording the $C_{10}H_{16}^+$ ($m/z$ 136) and $C_7H_9^+$ ($m/z$ 93) analyte ions). To facilitate identification of specific monoterpenes on the basis of the branching ratios of reactions (1) and (2), the analyte ion signal ratios [$m/z$ 81]/[$m/z$ 137] and [$m/z$ 93]/[$m/z$ 136] were determined under the same conditions of the *Profile 3* SIFT-MS instrument as used for the standard monoterpene mixtures. These branching ratios ($r$), given in Table 2, are discussed in Section 4.2.

The interaction of the primary ions with monoterpenes may be affected by the presence of neutral water molecules and thus by different sample humidity. Wang et al. (Wang et al., 2003) first reported this phenomonen when observing a change of the product ion signal ratio, $r$, in the reactions (see Section 3.2). For $H_3O^+$ reagent ions, this change was significant for β-pinene ($r$ reducing from 0.75 to 0.51), (R)-Limonene (0.45 to 0.34) and 3-carene (0.33 to 0.23). For the $NO^+$ reagent ion, a significant effect was observed only for α-pinene (0.32 to 0.08) and β-pinene (0.25 to 0.05). The decrease of $r$ can be explained by the

formation of hydrates of the reagent ions. It can be shown that the PA of monoterpenes is sufficient high to allow direct proton transfer from $H_3O^+.H_2O$ ions to the monoterpene molecules.

### 3.2 Analysis of the product ion intensity ratios

To facilitate assignment of the fast GC elution peaks to specific monoterpenes, mean fragment ion fractions $r_i = f_i/g_i =$ [$m/z$ 81]/[$m/z$ 137] (or for $NO^+$, $r_i = f_i/g_i =$ [$m/z$ 93]/[$m/z$ 136]) were calculated for each interval of retention times $t_1$ to $t_2$, as

the weighted mean of the product ion signal ratios $\overline{r_w}$:

$$\overline{r_w} = \sum_{i=t_1}^{t_2} w_i \frac{f_i}{g_i}; \; w_i = \frac{f_i + g_i}{\sum_{i=t_1}^{t_2} f_i + g_i}, \tag{3}$$

The weights ($w_i$) applied to each of several discreet measurements were based on the total signal count rates of both ions $f_i$ and $g_i$ in order to emphasise the area within the peak. Time intervals $t_1$ to $t_2$ were chosen for each isomer as the area of the chromatographic peak where the total ion signal was >10% of the peak value.

The quality of the ratio estimation was assessed from the variation of the $f_i/g_i$ ratio estimated as

$$\sigma_i^2 = Var(f/g) \approx \frac{\mu_f^2}{\mu_g^2}\left(\frac{\sigma_f^2}{\mu_f^2} + \frac{\sigma_g^2}{\mu_g^2}\right) = \frac{\mu_f^2}{\mu_g^2}\left(\frac{\lambda_f + \sigma_{bg_f}^2}{\mu_f^2} + \frac{\lambda_g + \sigma_{bg_g}^2}{\mu_g^2}\right), \tag{4}$$

where $\mu_f$ and $\mu_g$ represent intensities of the selected fragments and $\sigma_f^2$ and $\sigma_g^2$ are the variances of the $\mu_f$ and $\mu_g$ intensities estimated according to the Poisson distribution as the sum of distribution variance equal to the expected value $\lambda = \mu$ and background variance $\sigma_{bg}^2$ (Van Kempen and Van Vliet, 2000).

From this variation, the standard error of the weighted mean was calculated as:

$$\sigma_{\overline{r_w}} = \sqrt{\sum_{i=t_1}^{t_2} w_i{}^2 \sigma_i^2} \tag{5}$$

The weighted standard deviation of the $f_i/g_i$ ratios was also routinely calculated as:

$$s = \sqrt{\frac{\sum_{i=t_1}^{t_2} w_i \left(\frac{f_i}{g_i} - \overline{r_w}\right)^2}{1 - \sum_{i=t_1}^{t_2} w_i{}^2}} \tag{6}$$

### 3.3 Fast GC/SIFT-MS limits of detection and quantification

The total amount of eluting analyte, $C$, in each GC peak is determined by SIFT-MS from the area under the curve from the number density of the analyte molecules [M] (Španěl et al., 2006) in the flow tube recorded as a function of time, $t$, according to the equation:

$$C = \frac{1}{N_A} \int_0^{t_{max}} [M] S \, dt, \tag{7}$$

where $N_A$ is the Avogadro constant and $S$ is the constant volume flow rate of the sample and carrier gas mixture flowing into the SIFT-MS carrier gas as determined by the pumping speed of the SIFT-MS primary vacuum pump. Note that the flow rate of GC eluent gas does not enter this calculation and does not directly affect the determined amount of analyte expressed in nanomoles, nmol. [M] is calculated by the *Profile 3* software according to the SIFT-MS general method for the calculation of absolute trace gas concentrations from the reagent and product ion count rates, the reaction rate constants (see Table S1 in the Supplement) and the reaction time considering differential diffusion losses (see equation 15 in reference (Španěl et al., 2006).

The amount of neutral analyte (monoterpene) is proportional to its concertation [A] in sampled air and the sampled volume, $V$, given by the sampling flow rate (usually 3 sccm), and sampling time (1.8 to 12 s) as:

$$C = [A] \frac{V}{V_m}, \tag{8}$$

where $[V_m] = 24.0$ L mol$^{-1}$ is the molar volume of air at 293 K. Note that the sampled volume, $V$, calculated from the sampling flow rate and sampling time, changes with the column temperature as mentioned previously. The flow rate needs to be carefully determined by a direct flow measurement.

The limit of detection (LOD) was determined for α-pinene and (R)-limonene from analysis of a calibration curve as three times the standard error of the predicted intercept value divided by the slope of the calibration regression line (Graus et al., 2010). α-pinene and (R)-limonene were chosen as they have the lowest and the highest reaction rate constants for proton transfer (2.3 for α-pinene and 2.6 for (R)-limonene, in $10^{-9}$ cm$^3$ s$^{-1}$). For a reagent ion count rate was $10^6$ c s$^{-1}$ and a 12 seconds sampling interval, the LOD of the current setup was found to be 16.3 ppbv for α-pinene and 19.5 ppbv for (R)-limonene, using the column temperature 40 °C. For a column temperature 69 °C, the LOD for α-pinene decreased to 6.1 ppbv.

### 3.4 Reference chemicals and plant samples

All monoterpenes used in the experiments, viz. ((+)-α-pinene (98%), (+)-β-pinene (≥98.5% analytical standard), camphene (95%), myrcene (≥90% analytical standard), 3-carene (≥98.5% analytical standard), (+)-(R)-limonene (≥99.0% analytical

standard), α-terpinene (≥95%) and γ-terpinene (97 %), were purchased from Sigma-Aldrich. Individual monoterpene vapour standards and monoterpene vapour mixtures were prepared by the diffusion tube method (Thompson and Perry, 2009). Thus, for individual standards, about 5 µL of each monoterpene liquid was placed in a 2 mL vial closed by PTFE septum caps. Each vial was then penetrated with a diffusion tube (1/16" OD x 0.25 mm ID x 5 cm length PEEK capillary) and placed into a 15 mL

glass vial closed by a PTFE septum. The headspace of the 15 mL vial was sampled after stabilization (>30 minutes) of the concentration. The humidity of the headspace was typically 1.5% water vapour by volume as determined by SIFT-MS. For α-pinene, the vapour concentration was too high and thus it had to be reduced by placing only a much smaller amount of sample into the 2mL vial. For the mixture preparations, a similar approach was used in which several vials containing different monoterpene, penetrated by PEEK capillaries, were placed together into a 500 mL bottle. Note that the concentrations of the

individual isomers in the mixture were different due to the variations in the saturated vapour pressures of their liquids. The same mixture was used for $H_3O^+$ and $NO^+$ experiments with the MXT-1 column.

To demonstrate the applicability of the fast GC/SIFT-MS analyses to real samples, three different types of coniferous tree needles were prepared: Spruce (*Pincea punges*), Fir (*Abies concolor*) and Pine (*Pinus nigra*) (see Fig. S5 – S7 in the Supplement). For the first study using the MXT-1 column, the needle samples (0.26 g Spruce, 0.42 g Fir and 0.32 g Pine) were

collected in the urban area of Prague (June 2017) and placed into 10 mL vials from which the headspace was sampled 30 min after harvesting. For the later study using the MXT-Volatiles column, pine tree twigs were collected (June 2018) from the same trees (21.8 g Spruce, 21.4 g Fir and 20.6 g Pine). The exposed cuts of the twigs were sealed by wrapping parafilm around the cut. The samples were placed into a Nalophan bag of volume approximately one litre. During the analyses, the analytical laboratory was thermalized to the outdoor temperature (about 30 °C) to reduce thermal shock to the samples. In the laboratory,

only a scattered natural light was present.

## 4 Results and discussion

To investigate if the various monoterpenes in a mixture could be effectively distinguished using SIFT-MS enhanced by the fast GC pre-separation, eight common biogenic monoterpenes were investigated. Individual monoterpene standards were analysed first with both MXT-1 and MXT-Volatiles column to obtain the instrument response in terms of retention times and

product ion ratios using the two reagent ions $H_3O^+$ and $NO^+$. Then, the separation of monoterpenes was demonstrated through analysis of prepared monoterpene mixture. Separation of both GC columns was compared using isothermal GC at a temperature of 40 to 45 °C. The elution times of all studied monoterpenes were within 45 s of the total retention time for MXT-1 column and within 180 s for the MXT-Volatiles column. Using the information on the ratios of ion products for the $H_3O^+$ and $NO^+$ reactions together with the GC retention times, it was possible to identify the composition of a reference standard

mixture. Finally, the same procedure was used to analyse the leaf headspace of three coniferous samples to demonstrate the analysis of real samples.

## 4.1 Comparison of columns: MXT-1 vs. MXT-Volatiles

In the present experiment both columns were heated isothermally to approximately 40 °C selected to optimise temperature stability and chromatographic separation (see Fig. S4 in Supplement). For higher temperatures, the monoterpene chromatogram peaks coalesced while for lower temperatures a significant influence of the lab air temperature fluctuations was

apparent. However, even at these optimised conditions for the MXT-1 column, monoterpenes are not fully separated and thus, fast GC with the MXT-1 column alone (at 40 °C) provides only qualitative analysis.

The retention times determined from the chromatograms obtained for individual monoterpenes at 40 °C are given in Table 2, and further supported by Figure S2 in the Supplement. For the MXT-1 column, the apparent difference in retention times observed between the two reagent ions was probably caused by the temperature fluctuations of the column. Whilst the retention

times for individual monoterpenes are different, they are not sufficiently stable (fluctuate by > 1 s, see Table 2) in the present fast GC device for analyses based on retention time only to be reliable. A noticeable effect of ambient temperature on the rate of passive column cooling was observed resulting in changes of the column temperature profile and thus in variations of the monoterpene retention times. Therefore, for a longer column and a higher temperature it may be reduced. Use of the MXT-Volatiles column resulted in about five times longer retention times and better GC peaks separation at the same operational

conditions (flow rate, temperature and pressure) due to the higher efficiency of the 1.25 μm active phase (compared to 0.1 μm for MXT-1 column).

The quality of the separation can be increased by using hydrogen as a carrier gas and by a faster sample injection, as demonstrated by Materic et al. (Materić et al., 2015) with fastGC PTR-MS where complete separation of monoterpenes was achieved using the MXT-1 column. As observed for both columns, separation can be improved by decreasing the column

temperature (see Fig. 3 and Fig. S4 in the Supplement), however this increases the chromatogram width.

The performance of both the MXT-1 and MXT-Volatiles columns were compared by analyses of a gas mixture of eight monoterpenes. For the MXT-1 column, four characteristic GC peaks were identified for both reagent ions, marked as A, B, C and D with retention time of 17.6 s, 20.8 s, 26.3 s and ~30 s for $H_3O^+$, and 17.5 s, 20.7 s, 26.3 s and ~30 s for $NO^+$ (see Fig. 4). Based on the retention times obtained for individual monoterpenes (see Table 2 and Fig. S2 in the Supplement), peak A is due

to co-elution of α-pinene, camphene and myrcene. Peak B is due to the presence of β-pinene exclusively and peaks C and D are due to the remaining four monoterpenes, mainly 3-carene and (R)-Limonene. Note that the individual peak heights are influenced by the monoterpene saturated vapour pressures (see Table 2). Using the MXT-1 column under these conditions it is not possible to achieve separate GC peaks for individual monoterpenes; however qualitative analysis is possible.

The MXT-Volatiles column facilitates identification of all monoterpenes present in the mixture for a column temperature close

to room temperature (see Fig. 3). For the MXT-Volatiles tests, the sampling mode was extended to 12 s, representing the collection of approximately 0.6 mL of the monoterpene mixture headspace. At a column temperature 40 °C, the monoterpene peaks are well separated; however, α-pinene and camphene are likely to co-elute. It is interesting to note that the chromatogram (see Fig. S4 in the Supplement) changes with the temperature of the column and additional peaks appear at higher temperatures

probably as a result of the presence of different conformers. It thus seems that at the column temperature ~45 °C using 20 V heating voltage (see Fig. 4) in the mixture chromatogram, the small β-pinene is hidden behind the second camphene peak and the α-terpinene peak also disappears (see also the fragmentation analyses later in section 4.2).

**Table 2: Ratios of the $H_3O^+$ and $NO^+$ reaction product ion signals and the GC retention times, s, for the eight monoterpenes at columns temperature 40 °C. Also given are the saturated vapour pressures in Torr. The standard error of the fast GC $\overline{r_w}$ values for individual monoterpenes estimated by Eq. 5 is less than 5% (except 8.6% for camphene), overall less then ±0.02.**

| Compound | [m/z 81]/[m/z 137] | | [m/z 93]/[m/z 136] | | Retention time [s] | | |
| | $H_3O^+$ | | $NO^+$ | | $H_3O^+$ | $NO^+$ | $H_3O^+$ |
| *Saturated vapour pressure (Torr)* | Literature *Schoon* [a] *Wang* [b] | Results *Full scan fast GC MIM* | Literature *Schoon* [a] *Wang* [b] | Results *Full scan fast GC MIM* | MXT-1 | MXT-1 | MXT-Vol |
| α-pinene 4.75[e] | 0.45 | 0.67[c] | 0.05 | 0.16[c] | 16 | 14.7 | 72 |
| | 0.64 | 0.46[d] | 0.09 | 0.19[d] | | | |
| camphene 2.50[e] | 0.1 | 0.14[c] | 0 | - | 17 | 17.7 | 83 |
| | 0.16 | 0.16[d] | 0.01 | 0.03[d] | | | |
| β-pinene 2.93[e] | 0.52 | 0.61[c] | 0.03 | 0.12[c] | 20.4 | 22 | 106 |
| | 0.67 | 0.66[d] | 0.08 | 0.17[d] | | | |
| myrcene 2.09[f] | 0.44 | 0.72[c] | 0.36 | 0.72[c] | 18.5 | 17.8 | 134 |
| | 0.52 | 0.51[d] | 0.62 | 0.63[d] | | | |
| 3-carene 3.72[h] | 0.24 | 0.39[c] | 0.05 | 0.12[c] | 25.5 | 25.6 | 142 |
| | 0.32 | 0.35[d] | 0.1 | 0.15[d] | | | |
| α-terpinene 1.64[h], 1.66[i] | - | 0.14[c] | - | 0.01[c] | 27 | 25.1 | 157 |
| | 0.11 | 0.17[d] | | 0.01[d] | | | |
| (R)-limonene 1.98[g] | 0.30 | 0.43[c] | 0 | 0.03[c] | 27.5 | 31 | 170 |
| | 0.43 | 0.41[d] | 0.01 | 0.06[d] | | | |
| γ-terpinene 1.07[h], 0.7[j] | - | 0.18[c] | 0.08 | 0.08[c] | 40.4 | 32.5 | 184 |
| | 0.21 | 0.16[d] | 0.09 | 0.09[d] | | | |

[a] (Schoon et al., 2003); [b] (Wang et al., 2003); [c] Present result based on SIFT-MS measurement; [d] Present result based on fast GC/SIFT-MS measurement; saturated vapour pressures in Torr at 25 °C are according to [e] (Daubert, 1989), [f] (Haynes, 2014), [g] (Yaws, 1994), [h] (TGSC), [i] (Takasago, 2011), and at 20 °C according to [j] (ChemicalBook, 2016).

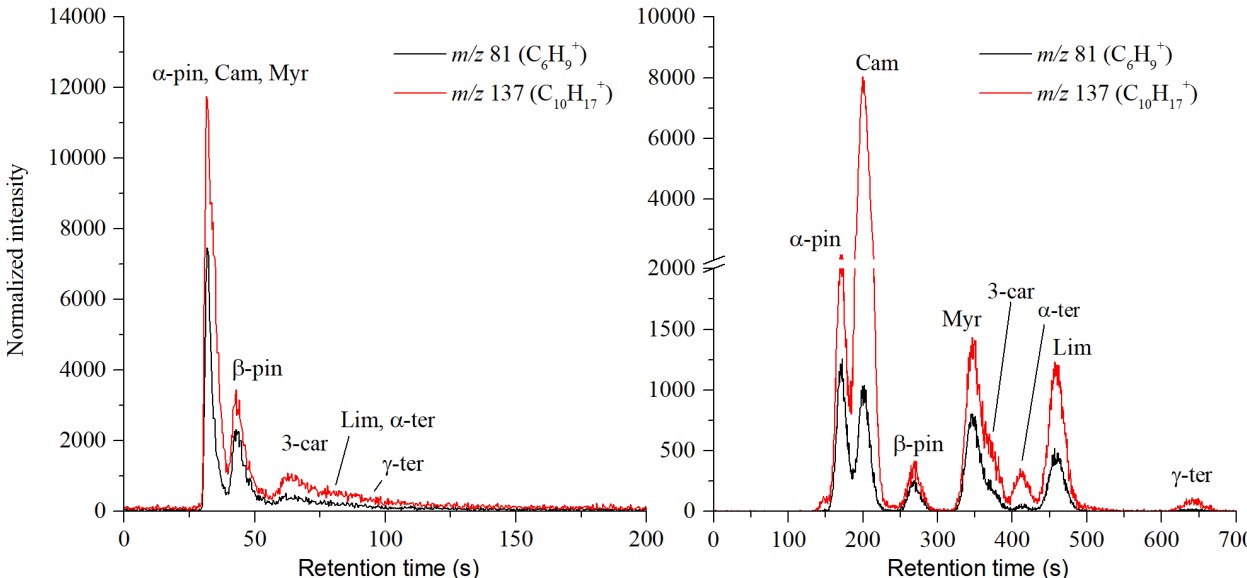

**Figure 3: Chromatograms of mixture of monoterpenes at room temperature obtained using the MXT-1 column (left) and the MXT-Volatiles column (right). Chromatogram peaks in the MXT-1 column are not fully separated, but separation takes less than 150 s compare to the 700 s required for the MXT-Volatiles column. The signal intensities are the analyte ion count rates normalized to a $H_3O^+$ reagent ion count rate of $10^6$ s$^{-1}$.**

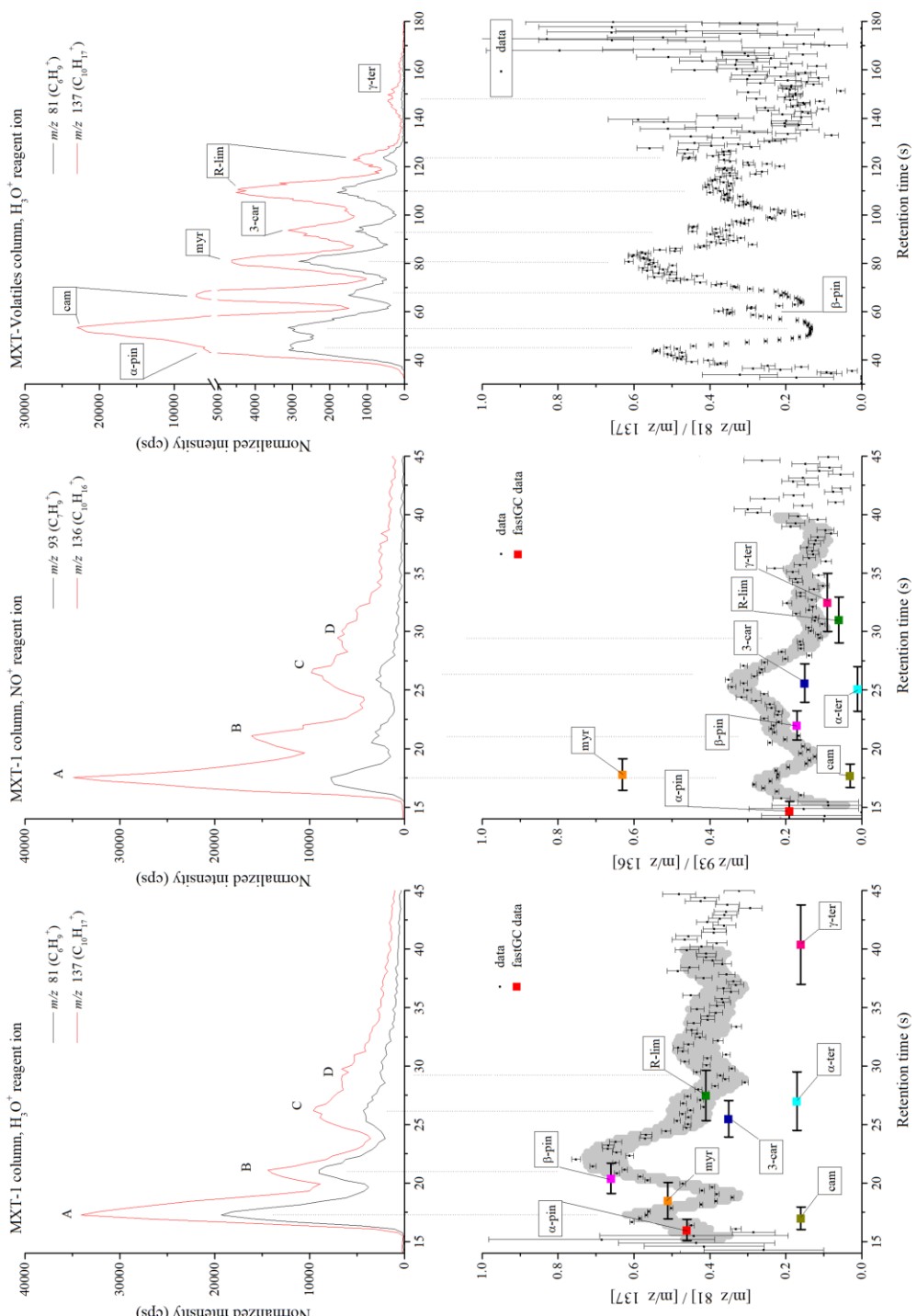

**Figure 4: Chromatograms of the mixture of monoterpenes (upper figures) measured by H₃O⁺ (left) and NO⁺ (right) reagent ions, obtained using the MXT-1 column. A, B, C, D represent characteristic peaks in the chromatogram. For each chromatogram, the product ion signal ratio $r_i$ is presented in the lower figures. The grey data background represents the calculated standard deviation of the data by Savizky-Golay smoothing between 15 s and 40 s. The position and value of the ratio for individual monoterpenes is based on the fast GC MXT-1 measurements presented in Table 1. Note that the retention times are determined by the fast GC conditions and do not depend on which SIFT-MS reagent ion is used. The signal intensities are the analyte ion count rates normalized to a reagent ion count rate of $10^6$ s⁻¹.**

## 4.2 Analysis of product ratio and use of the NO⁺ reagent ions

The inadequate separation of monoterpenes due to a short column or high temperature (as for the MXT-1 column) can be mitigated by the analysis of the product ion signal ratios $r_i$ (see Sec. 3.2) and additionally by using an additional reagent ion. It may be possible to improve identification of myrcene or camphene (often co-eluted with a-pinene) as well as of other monoterpenes by exploiting the different ion chemistry of the $NO^+$ reagent ions. These data in combination with $H_3O^+$ data allow identification of compounds on the basis of the ratios of four different product ions. The $NO^+$ reagent ions was used only for the MXT-1 column, because full separation of monoterpenes using $H_3O^+$ reagent ions was not achieved and thus retention time cannot be effectively used as a parameter for their identification. However, as will be shown, use of the $NO^+$ reagent ions brings additional benefits and thus it may be a valuable source of information even for fully separated chromatograms. Note that the retention times are determined by the fast GC conditions and do not depend on which SIFT-MS reagent ion is used (see Table 2).

The $\overline{r_w}$ values (see Table 2) obtained from the SIFT-MS FS data and the MIM data for the fast GC peaks for most of the isomers are in good agreement. However, the ratios obtained for α-pinene and myrcene are somewhat variable between the FS and MIM data and they also differ somewhat from the literature values (α-pinene from 0.45 to 0.67 for $H_3O^+$, myrcene from 0.44 to 0.72 for $H_3O^+$). This may be caused by different humidities of the samples, as discussed in Section 3.1., where it was seen that increase of humidity lower the $\overline{r_w}$ values. In the fast-GC setup, water retention time is much shorter than the retention time for monoterpenes; thus, water influence on the ion chemistry is negligible for most monoterpenes. Slightly affected can be α-pinene as it is the first one presented in the chromatogram. Therefore, only $\overline{r_w}$ values obtained using the fast GC are used for further study. The standard error of the fast GC $\overline{r_w}$ values for individual monoterpenes estimated by Eq. 5 (using the MXT-1 column) is less than 5% (except 8.6% for camphene) and is smaller than the observed variability between the analytical methods. The $\overline{r_w}$ values for MXT-Volatiles column were similar to those obtained with MXT-1 column, as expected.

Analysis of $\overline{r_w}$ values can be now used to improve identification of monoterpenes in standard mixtures. For the MXT-1 column, the $\overline{r_w}$ values for peaks A, B, C and D (see Fig. 4) were calculated as 0.49±0.09, 0.63±0.07, 0.45±0.04 and 0.40±0.05 respectively for $H_3O^+$ and as 0.21±0.05, 0.21±0.04, 0.27±0.06, 0.14±0.03 for $NO^+$. Based on these ratios (using fast GC data from Table 2), peak B could clearly be assigned as β-pinene. However, the remaining peaks contain several isomers and thus the $\overline{r_w}$ values do not provide unique identifications. Therefore, the variations in the dynamic profile of $r_i$ needed to be investigated to see if it can provide additional information. The time profile of $r_w$ in the chromatogram is shown in the bottom part of Fig. 4. To recognize trends in these data, Savizky-Golay smoothing (Savitzky and Golay, 1964) was used (second order polynomial across 10 data points, OriginPro 9.0 (OriginPro, OriginLab Corporation, Northampton, MA, USA, 2018). Also plotted (grey area in Fig. 4) is the standard deviation of the data points from the smoothed line in the interval of retention times from 15 s to 40 s. Note that this standard deviation is greater than the standard error of the data points, possibly due to a lower accuracy of data at the longer retention times. The standard deviation allows assessment of the significance of the changes in $r_i = f_i/g_i$.

According to the elution time, the first chromatographic peak A consist of three monoterpenes: α-pinene, camphene and myrcene. For the $H_3O^+$ reagent ions, the $\bar{r_w}$ value corresponds to both α-pinene and myrcene considering the $\bar{r_w}$ value for peak A (0.49) or $r_w$ close to the peak maxima (0.55–0.6). However, a more obvious difference between α-pinene and myrcene is observed using $NO^+$ reagent ions. The value of the weighted mean ratio for the peak A (0.21) is close to the ratio for α-pinene.

In the maxima of peak A, however, $r_w$ approaches the value of 0.3, which is close to the value expected for a combination of both these monoterpenes (0.32, considering the data from fast GC measurement and the vapour pressure in Table 2). For camphene, $r_w$ in the chromatograph did not reach the low values expected for both reagent ions. However, its presence is clearly visible as a dip in $r_w$ situated between the peaks A and B. In the absence of camphene, the ratio should linearly move to values characteristic of peak B without any dip. The depth of the dip does not reach the $r_w$ expected for camphene due to a

persistent tails of the peaks for both α-pinene and myrcene.

Peak B in the chromatograms is identified as β-pinene by its retention time. The $\bar{r_w}$ values for the $H_3O^+$ and $NO^+$ reagent ions are 0.63 and 0.21, respectively. The $r_w$ values are similar to $\bar{r_w}$ and slightly higher than to the fast GC standard values for β-pinene (see Table 2).

Peaks C and D are not clearly separated in the chromatogram. For the $H_3O^+$ reagent ions, the $\bar{r_w}$ value is similar for both peaks;

thus, the presence of (R)-limonene, 3-carene or α-terpenine is likely since the $\bar{r_w}$ values for the peaks C (0.45) and D (0.4) are comparable with the analyte signal ratios (see Table 2) for (R)-limonene and 3-carene. A lower $r_i$ for α-terpenine might be interpreted as a dip similar to that for camphene. However, the observed dip in $r_i$ at the D peak is not so statistically significant as is the dip for camphene, and the vapour pressure for both α- and γ-terpenine are lower than those for the other monoterpenes. Analysis of the C and D peaks using the $NO^+$ reagent ion shows a clearer difference between them. The calculated $\bar{r_w}$ for the

peak C (0.27) as well as the maximum $r_i$ (0.35) are, unexpectedly, much higher than for the remaining monoterpenes. This can be explained only by the influence of myrcene or by the presence of impurities in the form of an additional monoterpene in the mixture (for example, ocimeme has a high $r_i$ of 0.62 (Wang et al., 2003)). Amongst the eight monoterpenes, 3-carene has the highest $r_i$ within the retention time of peak C. The second peak D (0.14) can be then associated with (R)-limonene, which has a low $r_i$ (0.06) for $NO^+$ reagent ions, with some contribution by α-terpenine. The presence of γ-terpenine is not

apparent due to its low vapour pressure, but there may be some contribution in the D peak, but much smaller than the contribution by (R)-limonene.

To summarize, combining analyses using both $H_3O^+$ and $NO^+$ reagent ions with dynamic variations of $r_i$ allows the identification of α-pinene, camphene and myrcene in peak A  and by the presence of β-pinene only in peak B. Peak C is characterized as 3-carene and peak D as (R)-limonene and/or α-terpenine. γ-terpenine contributes only weakly due to its low

vapour pressure and has no recognisable response in the chromatogram compared to the remaining monoterpenes.

Analysis of the $\bar{r_w}$ values for the MXT-Volatiles column is simpler due to better separation of peaks. The value of $r_i$ clearly change for different monoterpenes, according to the expected $\bar{r_w}$ values for individual monoterpenes. The usefulness of the $r_i$ analysis for the MXT-Volatiles column can be observed in the analysis of β-pinene, which is featureless compared to that for

camphene. Camphene, additionally, produces a second chromatographic peak, which can be incorrectly associated with β-pinene. Analysis of the $r_i$ show values below 0.2 for both peak maxima, characteristic of camphene. The presence of β-pinene is visible as an increase of the $r_i$ value up to 0.4 at a retention time 60 s.

## 4.3 Tree samples investigation using the MXT-1 column

To test how the fast GC/SIFT-MS combination is applicable for analyses of real botanical samples, VOC emissions were analysed from three fresh coniferous tree needle samples: spruce, fir, and pine as shown in Fig. 5. The analytical MS obtained using $H_3O^+$ reagent ion are shown in Fig. S3 in the Supplement. Based on the results of the above GC data for standard monoterpene mixtures, the chromatograms were divided into three regions. The first region is characterized by the presence of α-pinene, camphene and myrcene between retention times of 12-18 s, the second region is characterized by the presence of

β-pinene with retention times between 18-25 s and the third region characterized is by presence of 3-carene and (R)-limonene with retention times between 25-40 s. The $\overline{r_w}$ values were calculated for the specific regions as follows:

- Spruce: The first region of the main peak, 0.35±0.07 ($H_3O^+$), 0.11±0.04 ($NO^+$). Note that the very low $\overline{r_w}$ for $NO^+$ indicates the absence of myrcene. The $\overline{r_w}$ value for $H_3O^+$ is lower than expected for β-pinene and higher than expected for camphene. Therefore, the first peak is mainly due to α-pinene, perhaps with a small amount of camphene. The

second region of the main peak, 0.31±0.07 ($H_3O^+$) and 0.09±0.08 ($NO^+$). $\overline{r_w}$ for $H_3O^+$ is lower than expected for β-pinene and higher than that for camphene The signal therefore belongs to the decay of α-pinene. The signal ratio 0.38±0.14 ($H_3O^+$), 0.14±0.12($NO^+$) in the third region indicates presence of (R)-limonene or 3-carene.

- Fir: The chromatogram shows two large peaks. The calculations of $\overline{r_w}$ for the first region (0.40±0.04 for $H_3O^+$, 0.14±0.04 for $NO^+$) and for the second region (0.56±0.04 for $H_3O^+$, 0.15±0.02 for $NO^+$) indicate the presence of both

α-pinene and β-pinene. The decreasing $\overline{r_w}$ for the $H_3O^+$ reagent ions in the last part (0.48±0.06 for $H_3O^+$, 0.19±0.05 for $NO^+$) indicates the presence of 3-carene .

- Pine: The chromatogram contains only one peak. $\overline{r_w}$ is stable for both reagent ions for all retention times (0.55±0.06 for $H_3O^+$, 0.21±0.05 for $NO^+$ for the first sector; 0.57±0.05 for $H_3O^+$, 0.22±0.04 for $NO^+$ for the second sector; 0.57±0.09 for $H_3O^+$, 0.22±0.10 for $NO^+$ for the third sector). Together with the retention time of the peak (16.4 s) this

certainly corresponds to α-pinene.

Concentrations of individual monoterpenes were calculated according to the procedure described in Section 3.3 for all selected regions. Calculation of monoterpene concentrations depends primarily on the individual reaction rate constants (see Table S1 in Supplement), which change from 2.3 to 2.6 for $H_3O^+$ and from 2.0 to 2.3 for $NO^+$ (in units of $10^{-9} cm^3 s^{-1}$). Incorrect identification of the monoterpene will thus lead to a maximum 20% error in the concentration calculation. According to the $\overline{r_w}$

values in selected regions, the most representative rate constant was adopted to calculated the monoterpene concentration in the selected region (see Table 3).

**Table 3: Calculated concentrations of monoterpenes (in ppmv and %) in the headspace over coniferous needles in selected regions of chromatograms obtained using MXT-1 column at column temperature 40 °C, using injection time 1.8 s and column flow 3 sccm. Rate constant used for the calculation of concentration in selected regions was chosen according to the $\overline{r_w}$ analysis.**

| Sample | Concentration (ppmv, %) | | | |
|---|---|---|---|---|
| | 12-18s | 18-25s | 25-40s | Sum 12-40s |
| Spruce ($H_3O^+$) | 11.0[A], **42%** | 9.0[A], **35%** | 5.2[R], 5.9[3], **23%** | 25.2 [A,R], 25.9 [A,3] |
| Spruce ($NO^+$) | 14.5[A], **50%** | 6.6[A], **23%** | 7.4[R], 7.7[3], **27%** | 28.5 [A,R], 28.8 [A,3] |
| Fir ($H_3O^+$) | 177[A], **32%** | 274[B], **49%** | 95[R], 107[3], **19%** | 546 [A,B,R], 558 [A,B,3] |
| Fir ($NO^+$) | 117[A], **31%** | 191[B], **51%** | 74[R], 77[3], **18%** | 372 [A,B,R], 375 [A,B,3] |
| Pine ($H_3O^+$) | 195[A], **55%** | 112[A], **31%** | 43[R], 49[3], **14%** | 350 [A,R], 356 [A,3] |
| Pine ($NO^+$) | 128[A], **48%** | 100[A], **37%** | 38[R], 41[3], **15%** | 266 [A,R], 269 [A,3] |

Calculations were performed using the reaction rate constants for [A] α-pinene, [B] β-pinene, [R] (R)-limonene or [3] 3-carene.

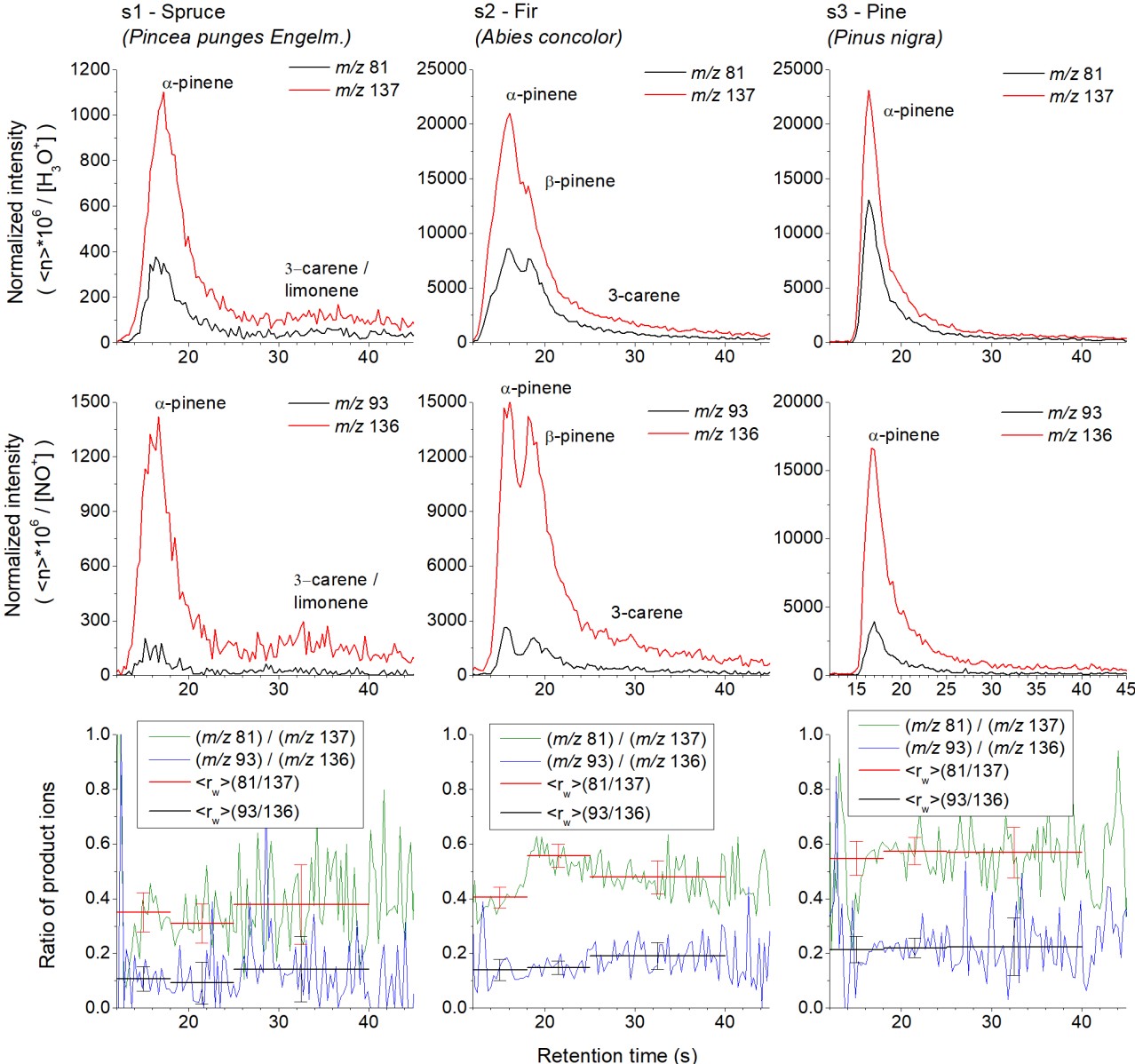

**Figure 5: Chromatograms derived using the product ions for the reactions of $H_3O^+$ (upper row) and $NO^+$ (lower row) reagent ions with monoterpenes obtained for the three investigated pine tree samples (s1, s2 and s3) using the MXT-1 column. The signal intensities are the analyte ion count rates normalized to a reagent ion count rate of $10^6$ s$^{-1}$. The black and red curves represent $C_6H_9^+$ (m/z 81) and $C_{10}H_{17}^+$ (m/z 137) product ions for $H_3O^+$ and $C_7H_9^+$ (m/z 93) and $C_{10}H_{16}^+$ (m/z 136) product ions for $NO^+$ reagent ions. The last row shows calculated ratios of product ions $r_i$ for both reagent ions (green and blue curves) and for peaks areas calculated $\overline{r_w}$ (red and black).**

**4.4 Tree samples analyses using the MXT-Volatiles column**

Similar experiments were conducted also using the MXT-Volatiles column, although on a different set of coniferous samples. The retention times for the individual monoterpenes were taken from the standard data obtained at the same column temperature (40 °C). The higher retention times of the MXT-Volatiles provides more accurate peaks identification than does the MXT-1 analysis. However, the different sample type resulted into a lower monoterpene concentration and thus the uncertainty of the $\overline{r_w}$ values significantly increased. The headspaces of the prepared tree needles were sampled for 6 s, representing a headspace volume of 0.3 mL. The chromatograms obtained for the spruce, fir and pine samples are shown in Fig. 6 and represent the means of analyte ion count rates from 5 consecutive runs normalized to a constant reagent ion count rate of $10^6$ s$^{-1}$.

- Spruce: In the chromatogram, four peaks were observed. The first peak with a retention time of 68 s corresponds to α-pinene with $\overline{r_w}$ of 0.60±0.16 for H$_3$O$^+$ and 0.24±0.15 for NO$^+$ reagent ions. The trailing edge of the first peak shows a decrease of $\overline{r_w}$ (0.29±0.11 for H$_3$O$^+$, 0.14±0.26 for NO$^+$) attributed to a small contribution by camphene. The second peak attributed to β-pinene, characterized by a retention time of 94 s with $\overline{r_w}$ of 1.05±0.59 for H$_3$O$^+$ and 0.50±0.15 for NO$^+$. The standard deviation in $r_w$ was unfortunately substantial. The position of the third peak is assign to myrcene. The $\overline{r_w}$ values (0.43±0.25 for H$_3$O$^+$, 0.41±0.54 for NO$^+$) were again imprecise due to the low intensity and do not fully agree with the unique $\overline{r_w}$ for myrcene (see Table 2). The observed weak peak could therefore be due to monoterpenes other than those eight included in Table 1. The last peak is associated with 3-carene with $\overline{r_w}$ as 0.48±0.27 for H$_3$O$^+$ and 0.16±0.39 for NO$^+$ reagent ions.

- Fir: In the chromatogram, three peaks are present where the first is due to both α-pinene and camphene. Transition of $\overline{r_w}$ from the left (0.57±0.21 for H$_3$O$^+$, 0.23±0.13 for NO$^+$) to the right (0.22±0.07 for H$_3$O$^+$, 0.04±0.04 for NO$^+$) part of the first peak is clearly visible on the Fig. 6 in the middle column. The first peak thus consists of two isomers. The second peak is attributed to β-pinene ($\overline{r_w}$ 0.80±0.21 for H$_3$O$^+$, 0.26±0.19 for NO$^+$) and the third peak is attributed to 3-carene ($\overline{r_w}$ 0.39±0.17 for H$_3$O$^+$, 0.15±0.27 for NO$^+$).

- Pine: The chromatogram shows three clear peaks due to α-pinene (0.73±0.13 for H$_3$O$^+$, 0.30±0.04 for NO$^+$), β-pinene (0.92±0.22 for H$_3$O$^+$, 0.26±0.13 for NO$^+$) and 3-carene (0.49±0.15 for H$_3$O$^+$, 0.13±0.15 for NO$^+$) with just a very small and statistically insignificant indication of camphene. The retention times for α-pinene, β-pinene and 3-carene were 69.6 s, 97 s and 141 s, respectively.

The concentrations of individual monoterpenes were calculated according to the procedure described in Section 3.3 based on the individual reaction rate constants (see Table S1 in the Supplement). Calculated monoterpene concentration are presented in Table 4.

**Table 4: Calculated concentrations of monoterpenes (in ppmv and %) in the headspace of coniferous twigs in selected regions of chromatogram obtained using the MXT-Volatiles column at a column temperature of 40 °C, using an injection time of 6 s and a column flow of 3 sccm.**

| Sample | Concentration (ppmv, **%**) | | | | |
|---|---|---|---|---|---|
| | α-pinene | Camphene | β-pinene | 3-carene | Sum |
| Spruce ($H_3O^+$) | 0.97, **46%** | 0.21, **10%** | 0.46, **22%** | 0.48, **22%** | 2.12 |
| Spruce ($NO^+$) | 0.74, **36%** | 0.26, **13%** | 0.56, **27%** | 0.49, **24%** | 2.05 |
| Fir ($H_3O^+$) | 2.51, **31%** | 1.46, **18%** | 2.9, **36%** | 1.17, **15%** | 8.04 |
| Fir ($NO^+$) | 1.97, **28%** | 1.29, **19%** | 2.80, **40%** | 0.88, **13%** | 6.94 |
| Pine ($H_3O^+$) | 15.5, **65%** | nd | 5.95, **25%** | 2.29, **10%** | 23.74 |
| Pine ($NO^+$) | 13.7, **65%** | nd | 5.45, **26%** | 1.83, **9%** | 20.98 |

nd – no data

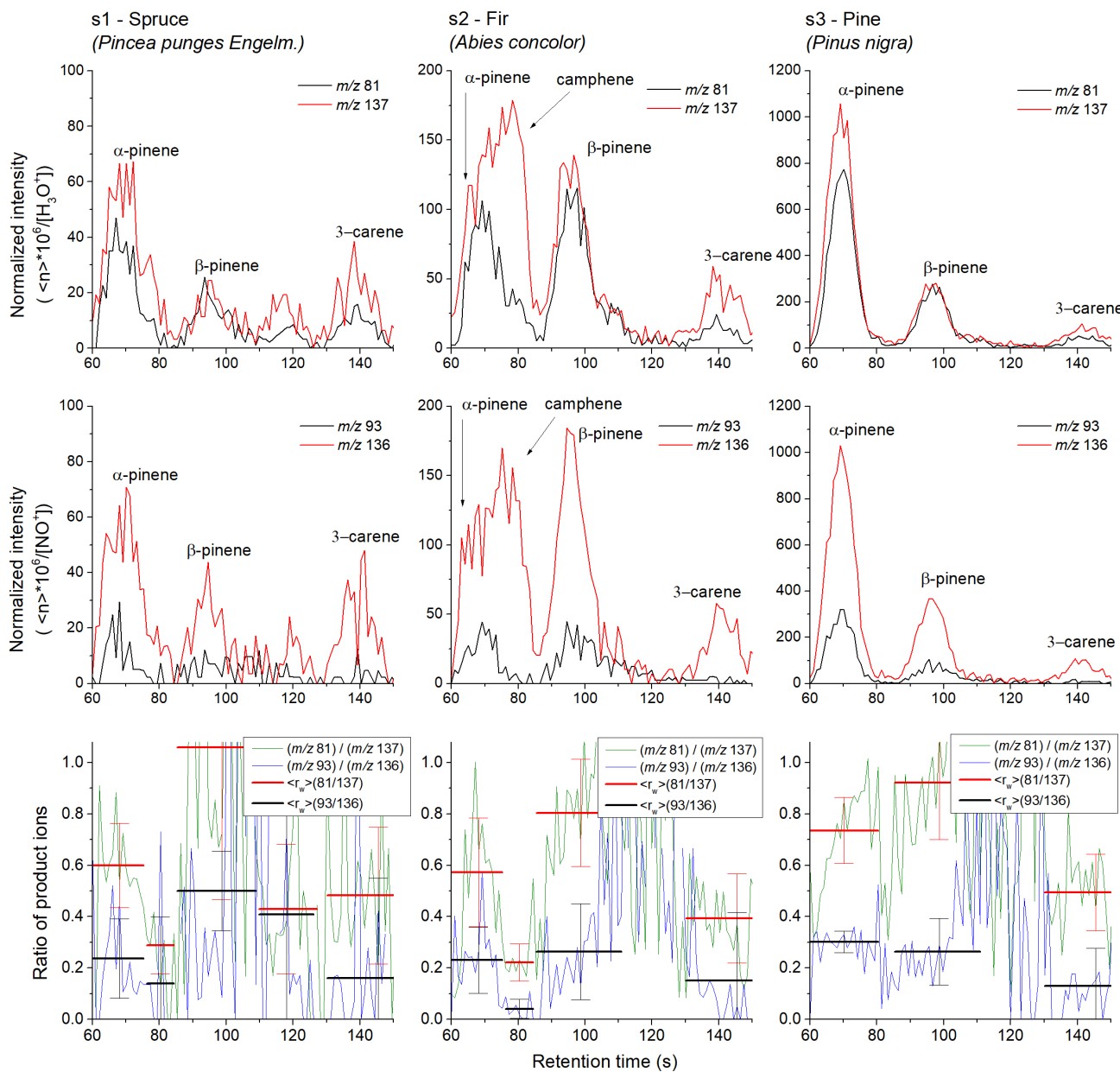

**Figure 6: SIFT-MS selected ion mode/fast GC/SIFT-MS chromatograms for monoterpene emissions from pine tree samples (s1, s2 and s3) obtained using the MXT-Volatiles column. The upper and lower rows were obtained using $H_3O^+$ and $NO^+$ reagent ions respectively. The signal intensities are the analyte ion count rates normalized to a reagent ion count rate of $10^6$ s$^{-1}$. The black and red curves stand for monitored ions $C_6H_9^+$ (m/z 81) and $C_{10}H_{17}^+$ (m/z 137) for $H_3O^+$ reagent ions and $C_7H_9^+$ (m/z 93) and $C_{10}H_{16}^+$ (m/z 136) for $NO^+$ reagent ions respectively. The last row shows calculated ratios of product ions $r_i$ for both reagent ions (green and blue curves) and for calculated peaks areas $\overline{r_w}$ (red and black). The signal intensities are the analyte ion count rates normalized to a reagent ion count rate of $10^6$ s$^{-1}$.**

## 4.5 Comparison of the tree samples analyses

Some differences are seen between the results from the MXT-1 and MXT-Volatiles columns. The most significant difference is the presence of a camphene peak in the fir sample headspace, and the presence of β-pinene and 3-carene in the pine sample headspace when the MXT-Volatiles column was used. However, samples were collected at different times of the year and the character of the samples was also different (only needles for MXT-1 and whole twigs for the MXT-Volatiles analyses). Different sample sources could also cause differences in monoterpene concentration (see Table 3 and 4).

Additionally, the recorded analyte ions may include interference by ions originating from other BVOCs emitted by the samples, especially when plants are physically damaged, since they emit so called „leaf aldehydes" such as 2-, and 3-hexenal (Tani et al., 2003). Whilst the reaction of 2-hexenal with $H_3O^+$ proceeds as a proton transfer forming a product ion at $m/z$ 99 (100 %), it has been found that reaction of cis-3-hexenal with $H_3O^+$ results in $H_2O$ elimination producing a dominant fragment at $m/z$ 81 (Španěl et al., 1997). If these interferences occur, they may eventually lead to the increase and to misinterpretation of the estimated $\overline{r_w}$ value. To avoid an overlap of 3-hexenal with monoterpenes, it is thus more reliable to use the product/analyte ion at $m/z$ 137 and exclude the $m/z$ 81 ion. Another possibility is to choose $NO^+$ as a precursor ion, where the product ions of 3-hexenal ($m/z$ 97, 69 and 74) do not overlap with those of monoterpenes ($m/z$ 92, 93 and 136) (Wang et al., 2003). Unfortunately, we did not carry out the fast GC analysis of 3-hexenal, so we do not know if it actually interfered with any of the detected monoterpene peaks.

## 4.6 Comparison with previous studies

The present experiments indicate that using the fast GC/SIFT-MS combination, it is possible to achieve analysis of monoterpene mixture. The estimated LOD are as follows: 16.3 ppbv for α-pinene and 19.5 ppbv for (R)-limonene, using the column temperature at 40 °C, and for the column temperature 69 °C, the LOD for α-pinene decreased to 6.1 ppbv. This is inferior to the previously described limit of the detection of up to 1-2 ppbv and full separation achieved by a fastGC-PTR-MS systems (Materić et al., 2015; Pallozzi et al., 2016). The higher LOD of the fast GC/SIFT-MS combination is due to the low flow rate of the sampling gas (~3 sccm) through the fast GC column, which is less than the commonly used 30 sccm. This could be resolved by using a wider column or by using multiple capillaries in parallel.

However, one clear advantage of SIFT-MS analyses is the ability to use three reagent ion which provide different analyte ions. This study has shown that the combination of the data from the two reagent ions, together with the analyses of the product ion signal ratios $r_i$, can improve the identification of monoterpenes, especially the identification of camphene and myrcene.

Importantly, it must be kept in mind, that monoterpenes are not the only BVOCs emitted by plants. The presence of 2-, and 3-hexenal, as already discussed in Section 4.5, can be problematical, but interference from this can be alleviated using $NO^+$ reagent ions. The same approach may be used to analyse other isomeric or isobaric molecules present in the environment. A further benefit of employing $NO^+$ reagent ions in atmospheric analysis is the quantification of isoprene, which when using $H_3O^+$ reagent ion mode suffers mass interference from product ions of other biogenic species, including furan, C5 aldehydes

and 2-methyl-3-buten-2-ol (Karl et al., 2012; Karl et al., 2014) as well as the second hydrate of methanol that is also emitted by plants (12% of global BVOC emissions) (Španěl et al., 1999). Another benefit of using SIFT-MS compared to other techniques is that calculation of VOC concentration in the sample depends only on the known physical constants, reaction rate constant and analyte ion abundance, so complicated calibration procedures are not required.

The results obtained for monoterpene composition in leaf headspace samples agree well with other published studies. Because the emission from plants depends on various physical parameters, here we only compare monoterpene composition. In a previous study (Mumm et al., 2004) of the volatiles emitted by *Pinus nigra* needles, 35 terpenoid compounds were identified, with the following being most abundant: α-pinene (45%), β-phellandrene (9%), limonene (8%), β-pinene (5%) and 3-carene (2%). Holzke et al. (2006) studied diurnal and seasonal variation of monoterpenes and sesquiterpenes from Scots pine. The

main monoterpene isomers they observed were α-pinene, β-pinene and 3-carene, which represented 90% of the total terpene emission. A similar study on monoterpene emissions from boreal Scots pine showed that the most abundant monoterpenes measured above the forest and from the canopy were α-pinene and 3-carene (Räisänen et al., 2009). Kainulainen et al. (Kainulainen et al., 1992) investigated the effect of drought and waterlogging stress on monoterpenes released by needles of *Picea abies* (spruce). In the controlled group, the most abundant monoterpenes were camphene (22%), limonene (14%), α-

pinene (9%) and myrcene (6%). In the emission from Southern and Central Sweden spruce (Janson, 1993) the following isomers were most abundant: α-pinene (60-70%), camphene (10%), limonene (10%) and 3-carene (4%).

Zavarin et al. (Zavarin et al., 1975) studied cortical oleoresin from *Abies concolor* (fir) were collected in 43 different localities in order to analyse their composition for the monoterpenoid fractions. They concluded that the production of camphene and 3-carene varied geographically. In the study of Pureswaran et al. (Pureswaran et al., 2004) they focused on quantitative variations

in monoterpenes from four species of conifers, concluding that the four species (Douglas-fir, Lodgepole pine, Interior spruce and Interior Fir) did not differ qualitatively but there were significant differences in their quantitative profiles. For example, Coastal Douglas fir needle samples contained 10% of α-pinene, 31% of Sabinene and 40% of β-pinene, and in samples of interior Douglas fir the most abundant isomers were bornyl acetate (26%), camphene (25%), α-pinene and β-pinene (both 15%).

In the present headspace study, we detected the presence of α-pinene, β-pinene, camphene and 3-carene, representing common emissions emitted from pine, spruce and fir samples. The present results thus agree with the usually reported composition of monoterpenes emitted from pine trees and their constituent parts.

**5 Summary and conclusions**

The addition of a fast GC pre-separation stage to SIFT-MS allows analyses of monoterpenes in mixtures at the expense of

some loss of sensitivity. The bespoke electrically heated fast GC systems constructed for this study achieved separation in less than 45 s for a 5 m MXT-1 column and less than 180 s for a 5 m MXT-Volatiles column held at 40 °C. However, due to the insufficient GC separation, the analysis was not accurately quantitative, but it can be improved using a longer GC column

operating at higher temperature. The identification of individual monoterpenes was aided by using information on the ratios of the product/analyte ion signals of both $H_3O^+$ and $NO^+$ reagent ions. It was shown that combining the SIFT-MS product ion ratios and the GC retention times, 7 of 8 monoterpenes were identified in a prepared mixture using the MXT-Volatiles column. To demonstrate the analytical value of this novel combination of fast GC with SIFT-MS, volatile emissions from spruce, fir

and pine samples were analysed. α-pinene was identified together with smaller amounts of β-pinene and 3-carene. A significant contribution of camphene was also observed in the fir sample headspace.

Due to their different OH reactivity, the ability to distinguish individual monoterpenes at high time resolution with fast GC/ SIFT-MS has the potential to improve the understanding of the contribution of individual monoterpenes in atmospheric chemistry processes such as the formation of tropospheric ozone and secondary organic aerosols.

A major limitation of the fast GC/SIFT-MS system described here is the relatively high LOD (~ 16 ppbv), which currently preclude its application in measurement of monoterpenes in typical ambient concentrations. An additional weakness of the current fast GC setup is its relatively poor temperature stability caused by a strong dependence on the laboratory ambient temperature. But this can surely be improved by active temperature feedback to control the column temperature. The flow rate through the 5 m long and 0.28 mm i.d. column was about ten times lower than the conventional flow rate used in direct SIFT-

MS analyses and this resulted in commensurate worsening of the LOD This could be resolved by using a wider column or by using multiple capillaries in parallel. A clear advantage of SIFT-MS is the ready availability of three different reagent ions to determine different fragmentation ratios for the same retention time to improve the identification of compounds.

**Data availability**

All data are available upon request from the corresponding author (Michal Lacko).

**Author contribution**

ML and NW crated experimental hardwere and provided experiments with the MXT-1 column, ML, KS and PP then provided experiments with the MXT-Volatiles column. PS and ML provdided data reatment and paper preparation.

**Competing interests**

The authors declare that they have no conflict of interest.

**Acknowledgment**

This project has received funding from the European Union's Horizon 2020 research and innovation programme under the Marie Skłodowska-Curie grant agreement No 674911. Also we gratefully acknowledge partial funding from The Czech

Science Foundation (GACR Project No. 17-13157Y). We would like to thank Professor David Smith for his advice and help in the preparation of the manuscript.

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
