# Peer review of "Addition of a fast GC to SIFT-MS for analysis of individual monoterpenes in mixtures"

_Atmospheric Measurement Techniques, 2019_

## Short Comment (SC1) · 8 Apr 2019

Very nice work! I have a couple of questions:

1) What is the length of the second column that was used ("MXT-Volatiles")? Was it also 5 meters?

2) Since the fast-GC part is so simple, would it be possible to enhance the time resolution of the measurement by using parallel fast-GC lines? Basically to inject gas pulses into parallel columns one after another and analyze them sequentially.

3) As I understand it, normally fast-GC is done using smaller inner diameter (0.15-0.18 mm or even smaller) columns that what you have used here (0.25 mm). Can you comment why you used 0.28 mm columns and would it be possible to use a smaller

[Figure]

I.D. column instead (to increase gas velocity)?

Please note that there is a typo on page 2, line 15: "which can affects" -> "which can affect".

---

## Referee Comment (RC1) · Anonymous Referee #3 · 14 Apr 2019

The authors present the use of soft ionization mass spectrometry (SIFT-MS) combined with a fast-GC system in order to achieve separation and identification of different monoterpenes. The capabilities of two different columns are discussed. Furthermore, the potential use of different ionization modes when operating the SIFT-MS in order to better separate the monoterpene mixtures is suggested as a method to improve separation for this type of systems. After following the revisions suggested below, the publication should be suitable for AMT.

Specific comments

In the "abstract" and "summary and conclusions" sections of the manuscript, the achievement of quantitative analysis is suggested. This is not supported though by the main text and is even discussed that it's not the case by the authors on page 17,

line 14. To my understanding, a quantitative analysis would provide ppb values of the individual monoterpenes together with their detection limits. On the contrary, only normalized intensity values are provided throughout the whole manuscript, for a mixture of monoterpenes that are not fully separated in the conditions used except in one case-study where the retention times are high (Fig. S3, 5V, retention time: 500s). It is therefore essential that the abstract and summary are re-written to avoid any misleading suggestion of quantification that overpromotes the presented work. The authors should work towards providing a more representative view of the manuscript that is related to the separation optimization of a monoterpene mixture using a low-resolution fast GC combined with the information obtained from differences in fragmentation patterns when using different ionization in the SIFT-MS.

There is only one point in the manuscript where the authors discuss the detection limits of their technique that are as high as 100 ppb (page 17, line 15). How was that calculated? Did the authors perform calibrations for the individual monoterpenes? Where could this technique be applied with this high detection limits? I would expect that the values used in this study are not applicable to ambient field measurements since they are higher than any ambient observations. Comparison of this technique to other fast GC techniques shows differences in the limit of detection by orders of magnitude (page 17, line 15). As discussed in section 4.5, this technique is, therefore, inferior to others but could still be useful for identifying monoterpenes based on fragmentation. This should be the main part of the abstract and conclusions sections. This should be further discussed in the manuscript, especially since the authors attempt to publish in an atmospheric measurement technique journal.

In order to obtain valuable information, the authors suggest that changing ionization in the SIFT-MS is recommended. This implies that in order to obtain valuable information relative to other techniques the GC-SIFT should run in both ionization modes. What would be the time needed to go through an H3O+ and a NO+ cycle? How much more is the time compared to other fast GC techniques that only run once and with

better resolution (page 17, line 16)? Overall, I would recommend that the value of this work and the comparison of this technique to others should be further discussed and emphasized throughout the manuscript.

Section 2 is hard to read and I would suggest restructuring. In the first sentence of the section the authors introduce Fig. 1 but this is not followed by a discussion of the figure, the instrument parts, and operation. On the contrary, they discuss the column options and operating details and then go through the temperature profiles. I would recommend the following structure: A. A discussion of the parts of the fast GC pre-separation system and the modes of operation with their details that are discussed in section 2.1 and page 4, line 15 to page 5, line 4, B. Operating details together with columns of choice and temperature profiles.

In section 3.1 a short discussion regarding the humidity dependences is presented that is not supported by any figure or graph. Was the humidity of the different samples measured? If so, shouldn't these values be provided in all figures, especially since the effects seem to be substantial? Furthermore, this paragraph and further discussion should be part of the results and discussions and not the section it is now.

Section 4.1 and 4.2 have an overlap of results and discussion that makes these sections hard to follow. I would recommend that the authors work towards restructuring these sections to a clearer presentation of the results that the table and figures promote followed by a detailed discussion, for each graph, for each column, and the comparison of the two columns. A characteristic example of the difficulty of the reader to follow the results and discussion is the title of section 4.2 that has little to do with what is discussed in it. Furthermore, please discuss why NO+ was not tested for the MTX-Volatiles column.

Technical comments

Title and manuscript: change "analyses" to "analysis" Page 1, line 19: change to "...to separate them in less than 180 s...". Page 2, line 15: change to "... which can affect

human health...". Page 2, line 29: correct to "fast GC-PTR-ToF-MS" and in general correct throughout the manuscript "fastGC" to fast GC". Page 2, line 33: change to "we report method development results aimed to...". Page 6, line 27: change to "are given in Table 1, and discussed in section 4". Page 6, line 28: This is hard to follow sentence. Rephrase. Page 6, line 30: Change to "reagent" Page 8, line 2: change to "saturation vapor pressures" Page 17, line 27: change citation style Page 17, line 17: Which results? What are the authors comparing here? Page 18, line 22-23: "... allows analysis of mixtures of monoterpenes in the air in short time periods..." Is that the case for ambient measurements in the detection limits of the system? Isn't this overpromoting the capabilities of the system? Table S1: It will be nice to add the m/z of detection.

---

## Referee Comment (RC2) · Anonymous Referee #2 · 30 Apr 2019

The paper "Addition of a fast GC to SIFT-MS for analyses of individual monoterpenes in mixtures" by Lacko at all, submitted to AMT is work with potential. The addition of FastGC to the SIFT-MS is described in details, acknowledging most of the difficulties that method development brings. However, I find several major issues which prevent me from recommending its publication in AMT.

First, the work is unnecessary long for the amount of information given. Two columns were compared, MXT-1 and MXT-Volatiles, but only the latter one gives acceptable separation. It is clear that MXT-1 is not suitable for this system (too fast separation, with not much control over the retention time). For any future version of the manuscript I suggest avoid the entire sections of MTX-1 column (perhaps it could be briefly mentioned in the supplementary data).

[Figure]

Second, the detection limit for this system is 100 ppb. Unfortunately, this is not close to the ambient levels of monoterpene concentrations or any plant chamber experiment loads. So, the relevance of this method is not within the scope of ATM, but rather in the fields where the technique can be used (monoterpene concentrations >100 ppb). Thus, I suggest to the authors to consider submitting these findings to a more suitable journal dealing with mass spectrometry techniques in general.

Minor comments: The manuscript in general needs more clarity: E.g. In the Abstract "the headspace of three conifer needle samples was analysed" it is not clear what do you mean here. The abstract should be clear and stand-alone. I believe you mean "needle samples of three conniver species"? P4 L20. "(1 to 8 s)" is it 1 or 8 s you used? Or this is a range you can set? Again, I had to search in the following text to understand this better. P8 L3-10. Not entirely clear enough. How did you enclose the plant branches? You mention temperature stress! But, how long it passed from the cutting? Did you use any light during the measurement?

---

## Referee Comment (RC3) · Anonymous Referee #1 · 7 May 2019

This work describes a GC-CIMS measurement technique developed to improve understanding of the composition of monoterpenes in the atmosphere which is an active area of interest in the atmospheric chemistry community due to key their roles in processes leading to formation of ozone and secondary organic aerosol (SOA) and is therefore highly relevant to the scope of AMT.

A series of experiments on individual standards of monoterpene isomers, monoterpene standard mixtures and the headspace of conifer foliage samples using a bespoke fast GC system coupled with a SIFT-MS is presented to demonstrate the potential application of fast GC-SIFT-MS for the separation and analysis of monoterpenes and other isomers in atmospheric and laboratory studies that is not currently achievable with SIFT-MS alone. The performance of two different GC columns in the fast GC SIFT-MS

system was assessed - a generic (MXT-1) GC column and an application specific GC column (MXT-Volatiles). In addition, two reagent ions (NO+, H3O+) were used in the SIFT-MS system to aid in compound identification.

This work represents one of the first, if not the first, reported trial of a fast GC coupled with an SIFT-MS system which has a considerable user group worldwide. As noted in the manuscript introduction, this is an area of active development with previous papers describing fast GC coupled with other chemical ionization mass spectrometry (CIMS) systems, in particular PTR-MS (Materic et al 2015, Pallozzi et al 2016). Given the similarities between SIFT-MS and PTR-MS it could be considered that this paper does not represent a substantially novel development.

The original contributions to atmospheric measurement practice are:

1) The comparison of two GC columns - a generic (MXT-1) GC column (as used in previous fast-GC and GC-PTR-MS studies) and an application specific GC column (MXT-Volatiles) – this has relevance to the wider fast GC applications (SIFT-MS, PTR-MS, other CIMS, fast GC-FID...) in which MXT-1 column has been used.

2) The first reported use of NO+ reagent ions in a fast GC - CIMS set-up.

However, additional additions/revisions are required for substantial conclusions to be reached regarding the performance and potential applications of fast-GC-SIFT-MS for quantification of monoterpene isomers. Specifically, more quantitative information is required on the detection limits, sensitivity and procedures for the quantification of species concentrations- see specific comments below.

Specific comments

Detection limit - p 17 Line 15 states "The present experiments indicate that using the fast GC-SIFT-MS combination, it is possible to achieve only qualitative analysis of the monoterpene mixture with a limit of the detection of about 100 ppb." Detection limits of 100 ppb is a major limitation for the application of fastGC-SIFTMS to measurements

of individual monoterpenes in ambient air where concentrations are typically orders of magnitude lower (1 -10 ppb). The manuscript must include descriptions of:

1) How the stated detection limit of ∼100 ppb was determined?

2) Why is this detection limit so high?

3) Potential improvements to the instrumental set-up that would reduce the detection limit to a range that would allow its application to measurements of ambient air (< 1 ppb).

Without these additions the application of this measurement technique for atmospheric measurements is limited making the relevance of this work to AMT highly questionable.

Quantification - The abstract, p 1 Line 18 states "Thus, it is possible to quantify components of a monoterpene mixture in less than 45 s by the MXT-1 column and to separate them in less 180 s by the MXT Volatiles column."

Concentrations of monoterpenes are not quantified in this work and this claim is contradicted in the text p 17 Line 15 (as shown above) "it is possible to achieve only qualitative analysis of the monoterpene mixture". There are other similar contradictory statements in the manuscript which must be addressed.

Calibration – What is the sensitivity of this method? Was the system calibrated with certified gas standards containing one or more monoterpenes, and an empirical calibration factor determined?

Absolute quantification - In lieu of an empirical calibration factor, the well-defined conditions in the SIFT-MS permit calculation of the concentrations of monoterpenes based on the raw signals of reagent and analyte ions (ie [m/z 137] as defined in section 3.2 of the manuscript), known reaction rates, and branching ratiosand instrument parameters as described in the SIFT-MS literature (e.g. Smith and Spanel 2005, Mass Spectrom. Reviews, 24, 661 – 700).

[Figure]

Direct measurement via SIFT-MS - Was direct quantification via SIFT-MS (without GC column) performed? Few comparisons of NO+ and H3O+ measurements of monoterpenes are available in the published literature and would be a valuable contribution.

Both the detection limit and the sensitivity of the method are critical to understanding the application of this method for measurements of monoterpenes in the atmosphere and in laboratory studies. Neither are adequately described here making the relevance of this work to AMT highly questionable.

Relative abundance - In lieu of quantitative determination of individual monoterpene isomers, can the peak areas be used to estimate the relative abundance of each monoterpene species in the samples (mixtures and leaf headspace samples) ? Understanding the rel. abundance of monoterpenes is key to determining accurate calibration factors (see deGouw et al. (2003) JGR-Atmospheres 108, D21), and more importantly understanding the OH reactivity of BVOC dominated atmospheres. Suggest including NO+ and H3O+ reaction rates in Table 1 to demonstrate the importance of understanding the monoterpene composition to the accuracy of CIMS monoterpene measurements based on a single m/z, and adding a table of OH and O3 reaction rates for each monoterpene isomer identified and their relative abundance in leaf samples as well as some discussion regarding the potential contribution of different monoterpenes in the oxidation budgets of atmospheres dominated by emissions from these plant species. Overall, the measurement system and its operation are sufficiently explained however, inadequate information of the performance of this method in terms of detection limit and sensitivity are provided and potential future developments to improve performance are not adequately covered. Without this additional information the manuscript does not provide a substantial enough contribution to development of atmospheric measurement techniques for publication in AMT.

A key issue with CIMS instruments such as SIFT-MS and PTR-MS is essentially we know how much there is but we don't know what it is? Adding pre-separation techniques attempts to overcome this however, the data presented in this paper essentially

reverses the challenge- we know what there is but not how much ? The manuscript requires a clear procedure for the quantification of monoterpene concentrations and/or the relative abundance of monoterpene isomers from the raw data in order to demonstrate the usefulness of this method over direct measurements with SIFT-MS. Quantification has been demonstrated in related instruments (Jones et al 2014, Materic et al 2015, Pallozzi et al 2016).and it is unclear why it was not part of this work.

If these additions/revisions can be made, the following technical comments should also be considered.

Technical comments

Whole manuscript– replace SCI-MS with CIMS, the term chemical ionisation mass spectrometry (CIMS) is an established mass spectrometry term for analytical systems including SIFT-MS, PTR-MS etc

Abstract p1 line 18, change "quantify" to "qualitatively identify"

Abstract – add a couple of sentences at the end -what is the practical significance of this work? what is the theoretical significance?

P2 line 3, change "The analytical ion-molecule reactions" to " The chemical ionisation reactions"

P2, line 13, suggest addition of a new paragraph discussing the fact that due to issues with stability of monoterpene mixtures in certified gas standards, CIMS instruments employed in ambient air studies are often calibrated with certified gas standards containing only one or two monoterpenes, (typically a-pinene). However the instrument response differs between isomers due to differences in their ionization reaction rates and branching ratios. To determine an accurate (weighted) instrument sensitivity value for monoterpenes, the relative abundance of monoterpene isomers must be known (see deGouw et al. (2003) JGR-Atmospheres 108, D21).

P2 paragraph lines 13 – 21 – these concepts need to be re-visited in discussion and

summary to demonstrate the usefulness of these techniques.

P2 line 21, move these two sentences into subsequent paragraph "Gas chromatography mass spectrometry (GC-MS) coupled with pre-concentration techniques has been developed to successfully identify and quantify different atmospheric monoterpenes (Janson, 1993; Räisänen et al., 2009; Song et al., 2015). However, the requirements of pre-concentration and long cycle time (more than 1h) are obviously unsuitable for real-time measurements."

P4, "It is interesting to note that the flow of sampled air, established by the pressure difference between ambient atmosphere and the low pressure of the SIFT-MS flow tube, changes with the column temperature due to the variation of the dynamic viscosity of the air (see Fig. 2)." – Does this affect flow tube residence time (reaction time, t) important in SIFT-MS quantification calculations?

P4, line 16, Can measurements by the SIFT-MS when the GC set-up is in "normal mode" be considered an instrument zero (SIFT-MS instrument background)? Can you use this data to calculate the detection limit and subtract from "sampling mode" measurements?

P5, line 16- "Sampling was repeated several times to improve sensitivity." No data for sensitivity is presented.

P5 Section 3- insert details on the time it takes to switch between reagent ions and to achieve stable ion signals- this is crucial if NO+ and H3O+ are to be used for compound identification. What was the intensity and purity of the reagent ion signals?

P7 insert section (after section 3.2) describing quantification procedure (as discussed in specific comments above) either using empirically derived calibration factors or via absolute quantification procedure based on [m/z 137] for H3O+ mode; and [m/z 136] for NO+ mode.

P8 Section 4.1 Comparison of columns: MXT-1 vs MXT-volatiles. The comparison of

these two columns is valid given the use of the MXT-1 column in related instruments presented in the published literature (Jone et al 2014, Pallozzi et al 2016, Materic et al 2015 etc).

P8 paragraph line 12 -18 – Your approach needs to be more clearly articulated – for instance, firstly the instrument response to individual monoterpene species, in terms of retention time, and product ion ratios, was characterized via analysis of a series of prepared standards with both the MXT1 and MXT volatile columns and when H3O+ and NO+ were employed as the primary reagent ion in the SIFT-MS. Secondly, the separation of monoterpenes isomers using two columns, and the two reagent ions (NO+, H3O+) was demonstrated through analysis of prepared mixtures containing 8 monoterpenes. Lastly, the application of the GC-SIFT-MS for the separation (and quantification?) of monoterpene isomers in a real-world analysis is presented in a series of leaf headspace analyses.

Section 3.3, Note it is unclear whether the same individual standards and mixtures of monoterpene were analysed by both NO+ and H3O+ in the same analysis runs?

P8 line 22 – "Whilst the retention times for individual monoterpenes are different, they are not sufficiently stable (fluctuate by > 1 s, see Table 1) in the present fast GC device for analyses based on retention time only to be reliable." Suggested improvements to instrument design?

P8 line 28, the following statement is unclear "the peak shapes cannot be compared directly but the peak width (FWHM) increased only two times for the MXT-Volatiles column". Also define FWHM.

P9, Table 1 – add columns for reaction rates of monoterpenes with NO+ and H3O+ - consider landscape page layout (see comment above re Relative Abundance)

P11 Section 4.1 – Discussion of response to individual monoterpene standards. Insert Figure S2 and a corresponding plot for the MXT-volatiles column into section 4.1.

These are very helpful when interpreting subsequent Figures 3 and 4. What conclusions can be reached from the tests of individual monoterpene standards – based on these tests what peaks are likely to co-elute, and what peaks are likely to be able to be separated in analysis of an unknown mixture? These tests provide the fundamental information for interpretation of the data from mixtures and leaf samples and should be included in the main text.

P10 line 10 "As observed for both columns, separation can be improved by decreasing the column temperature (see Fig. S3 in the Supplement), however this may increase the chromatogram width and thus decrease the sensitivity of the technique. Additional sensitivity can be achieved by increasing the injection time, which will, however, increase the peak width." – this discussion is not quantitative, no explicit sensitivity data is presented.

The discussion in Section 4.1 regarding analysis of mixtures needs to be restructured.:

1) provide a direct comparison between MXT-1 and MXT-volatiles at the same conditions. ($\sim$40- 45C ). Figures 3 and 4 – Figure 3 is actually a comparison of H3O+ and NO+ and the data from the MXT1 and MXT-volatiles column are not compared side-by-side. Format a page in landscape orientation, combine figures 3 and 4 (three panels) and present them in a compatible format (ie same formatting and labelling etc).

2)Discuss challenges and potential improvements ie stability in retention times, improved separation via decreasing column temp, improved sensitivity by increasing injection times.

3)Present MXT-volatiles column data under optimized conditions – ie "The MXT-Volatiles column facilitates identification of all monoterpenes present in the mixture for temperatures close to room temperature (see Fig. S3 in the Supplement)." – the top panel in the S3 plot is key to demonstrating the achievable separation of the MXT-volatiles column - move it from the supplement to the main body. The additional species identifiable using this technique compared to the MXT-1 set-up need to be more clearly

summarised.

P12 Paragraph lines 8 – 17- needs to be moved to later in the discussion or into section 4.5 to show that aside from potentially better selectivity other co-benefits of employing the NO+ reagent ion in CIMS measurements of BVOCs, in particular in measurements of isoprene (See Karl et al 2012 ACP 12:11877 – 11884, and Karl et al 2014 Int J. Mass Spectrom. 365-366:15-19). There are many more species which interfere with quantification of isoprene in H3O+ reagent ion mode such as furan, 2,3,2-MBO, C5 aldehydes.

P12 line 19 " However, the ratios obtained for $\alpha$-pinene and myrcene are somewhat variable between the FS and MIM data and they also differ somewhat from the literature values." – be quantitative ie state % variability. Is the variability a result of changes in the reagent ion intensity (consider using normalised intensity), or composition (eg % reagent ion impurities of H3O+(H2O), O2+, NO+)?

P14 Section 4.3 –For this method to be useful in atmospheric research the concentrations of monoterpene isomers or an estimate of their relative abundance must be quantified from the data and presented here and section 4.4(see specific comments above re quantification).

P14 Section 4.4- be consistent – use dot point format as for previous section. Why is the data from non-optimized conditions (40C) presented? Was the analysis done at the optimal temperature (5V) for separation? If so, should be presented.

P14 line 14, " The signal increase in the third region may indicates trace presence of (R)-(+)-limonene." – the m/z81 signal or the ion intensity?- not clear.

P15 Section 4.4- need to state that similar experiments but on a different series of conifer samples were also conducted using the MXT-volatiles column.

P15 Figure 5- consistent units (normalised intensity) should be used for all figures (3-6), label peaks in both H3O+ and NO+ chromatograms (both Fig 5&6). Query the signal

to noise ratio of some of the identified peaks e.g. H3O+ spruce 3-carene / limonene peak. Re-iterates importance of quantifying method LoD.

P17 Section 4.4 – This section should conclude with a table of the relative abundance of each monoterpene isomer in the leaf samples and their reaction rates with OH and O3 with associated discussion.

P17 Section 4.5 – "The present experiments indicate that using the fast GC-SIFT-MS combination, it is possible to achieve only qualitative analysis of the monoterpene mixture with a limit of the detection of about 100 ppb. This is inferior to the previously described fastGC-PTR-MS systems (Materić et al., 2015; Pallozzi et al., 2016), which achieved full separation with limit of the detection up to 1-2 ppt." – list the reasons for the difference in performance and potential future developments of the GC-SIFT-MS method to improve performance. This statement must be addressed in more detail as these significant limitations preclude the application of this method to ambient studies and make the inclusion of this work in AMT questionable.

P17 line 17 – start new paragraph at "However, one advantage of SIFT-MS is the facility to use two reagent ions, and the analysis of product ion ratios provides additional information. Thus, the combination of the data from the two reagent ions together with the analyses of the product ion signal ratios ri can be shown to improve the identification of monoterpenes." – be more specific, what additional compounds were identified using the reagent ion chemistry. Suggest insert discussion from 4.2, on usefulness of NO+ reagent ion for identification of other BVOCs here. As a side note, switchable reagent capability has been developed for PTR-MS and other CIMS and is not unique to SIFT-MS.

P17 line 20 – "The results obtained from the present study agree well with the literature reports." Be more specific, suggest – the results obtained from the analysis of leaf headspace samples agree well other studies in the published literature. Suggest authors present comparisons by tree species as a table with following columnsplant species name; monoterpenes identified; rel. abundance where available; measurement method; time resolution; and where available: LoD & sensitivity; and literature reference. Focus discussion on number and rel. abundance of monoterpenes identified and the methods used, not on geographical variability or variability between species beyond the scope of this work. What is the potential advantage of this method over others? Time resolution?

P18 Section 5- "A new method has been developed that allows quantitative analyses of individual monoterpenes in mixtures using SIFT-MS enhanced by chromatographic pre-separation." As previously stated this is not correct and contradicts the first line of the previous section (4.5) "The present experiments indicate that using the fast GC-SIFT-MS combination, it is possible to achieve only qualitative analysis of the monoterpene mixture with a limit of the detection of about 100 ppb."

P18 line 16 start new paragraph at "A weakness of the current fast GC setup is the relatively poor temperature stability caused by a strong dependence on the laboratory ambient temperature...."

P18 line 18 "It has been shown that a clear advantage of SIFT-MS is the facility to use different reagent ions and to utilize the ratios of the specific product ions of their reactions with the various monoterpene isomers at the same retention time to improve the identification of the monoterpenes." Belongs in previous paragraph (P18, line 10).

P18 line 23 – "This novel idea of a fast GC-SIFT-MS combination could broaden the application of SIFT-MS to in situ trace gas analyses of complex mixtures such as ambient air and exhaled breath.". There are several issues with this statement:

1) SIFT-MS is already used for in situ ambient air and breath analysis- this technique GC-SIFTMS does not broaden its application. The practical significance of this work is that it aims to address the challenge of quantifying isomers in CIMS measurements of complex mixtures.

2) Also, need to preface this statement "With improved limits of detection and sensitivity, this novel fastGC-SIFT-MS could. . . . . . .." currently its application in ambient air analysis is limited due to high LoD and lack of data about its sensitivity.

What is the theoretical significance of this work- what will an improved understanding of the complex mixture of monoterpenes contribute to our understanding of atmospheric chemistry? Ie estimates of total OH reactivity etc.

---

## Author Comment (AC1) · 15 May 2019

Markus, thank you for your kind comment on our paper. Answers to your questions are below: 1) The length of both columns (MXT-1 and MXT-Volatiles) was identical, i.e. 5 meters. We will clarify this in the revised paper. 2) This is a very interesting suggestion. Sequential analysis, could be used to improve the time resolution of GC analysis down to several seconds. However, the hardware would have to be much more complicated and possibly artefacts due to the Nyquist Theorem could occur. Additionally, the idea of several parallel lines could simply be used to improve sensitivity by increasing the total flow of a smple whilst keeping the same quality of separation. 3) We have chosen the metallic column following the previous fast GC PTR-MS applications as it can be heated ohmically. Also the decreased I.D. would lead to a smaller flow rate limiting

sensitivity.

Thank you also for spotting the typo, we will correct it.

On behalf of the author team,

Patrik.

---

## Author Comment (AC2) · 15 May 2019

Thank you for the detailed and extensive commentary on our paper. It is absolutely true that quantification is an important aspect of analysis. As result of these comments we have decided to carry out additional experiments focused on validation of quantification and determination of the limits of detection, LOD, achievable by the fast GC - SIFT-MS combination.

Concerning the SCIMS and CIMS distinction we will consult our colleagues in the IM-PACT project from which this study is funded about their opinions, as we consider the SCIMS to be recognized as an distinct emerging family of techniques in biological, food and environmental analyses and it would be good to keep a consistent acronym also

in atmospheric measurements.

---

## Author Comment (AC3) · 15 May 2019

Thank you for the comments. We appreciate that 100 ppbv detection limit is of no use for atmospheric measurements, the manuscript as it was presented for discussion was focused on the principles of product ion ratios and of separation. As result of the anonymous comments, we will do more experiments and improve the sensitivity of the device and characterize LOD using calibration mixtures. This will be included in the revised version of the paper.

---

## Author Comment (AC4) · 15 May 2019

Thank you for constructive comments. We will certainly discuss quantification in the revised manuscript and also carry out additional experiments to obtain calibration curves from which the limits of detection can be calculated.

The detailed response to all referee comments will be provided with the revised manuscript after obtaining the results of additional experiments.

---

## Author Response (AR1)

Markus Metsälä

markus.metsala@helsinki.fi

Received and published: 8 April 2019

Very nice work! I have a couple of questions:

1) What is the length of the second column that was used ("MXT-Volatiles")? Was it also 5 meters?

The length of both used columns (MXT-1 and MXT-Volatiles) was 5 meters. This was clarified in the revised manuscript.

2) Since the fast-GC part is so simple, would it be possible to enhance the time resolution of the measurement by using parallel fast-GC lines? Basically to inject gas pulses into parallel columns one after another and analyze them sequentially.

This is a very interesting suggestion. Sequential analysis, could be used to improve the time resolution of GC analysis down to several seconds. However, the hardware would have to be much more complicated and possibly artefacts due to the Nyquist Theorem could occur. Additionally, the idea of several parallel lines could simply be used to improve sensitivity by increasing the total flow of a sample whilest keeping the same quality of separation. This idea was added to the conclusions as indication for further work.

3) As I understand it, normally fast-GC is done using smaller inner diameter (0.15- 0.18 mm or even smaller) columns that what you have used here (0.25 mm). Can you comment why you used 0.28 mm columns and would it be possible to use a smaller I.D. column instead (to increase gas velocity)?

We have chosen the metallic column following the previous fast GC PTR-MS applications as it can be heated ohmically. Also the decreased I.D. would lead to a smaller flow rate limiting sensitivity. We have proposed in conclusions to use even wider column for increased flow rate.

Please note that there is a typo on page 2, line 15: "which can affects" -> "which can affect".

Thank you. We have corrected these.

The authors present the use of soft ionization mass spectrometry (SIFT-MS) combined with a fast-GC system in order to achieve separation and identification of different monoterpenes. The capabilities of two different columns are discussed. Furthermore, the potential use of different ionization modes when operating the SIFT-MS in order to better separate the monoterpene mixtures is suggested as a method to improve separation for this type of systems. After following the revisions suggested below, the publication should be suitable for AMT.

**Specific comments**

In the "abstract" and "summary and conclusions" sections of the manuscript, the achievement of quantitative analysis is suggested. This is not supported though by the main text and is even discussed that it's not the case by the authors on page 17, line 14. To my understanding, a quantitative analysis would provide ppb values of the individual monoterpenes together with their detection limits. On the contrary, only normalized intensity values are provided throughout the whole manuscript, for a mixture of monoterpenes that are not fully separated in the conditions used except in one case-study where the retention times are high (Fig. S3, 5V, retention time: 500s). It is therefore essential that the abstract and summary are re-written to avoid any misleading suggestion of quantification that overpromotes the presented work. The authors should work towards providing a more representative view of the manuscript that is related to the separation optimization of a monoterpene mixture using a low-resolution fast GC combined with the information obtained from differences in fragmentation patterns when using different ionization in the SIFT-MS.

**This is an important comment we have thus extended the study by including quantitative method of calculation and by obtaining calibration curves from which actual LODs were calculated and these are given in the revised manuscript.**

There is only one point in the manuscript where the authors discuss the detection limits of their technique that are as high as 100 ppb (page 17, line 15). How was that calculated? Did the authors perform calibrations for the individual monoterpenes? Where could this technique be applied with this high detection limits? I would expect that the values used in this study are not applicable to ambient field measurements since they are higher than any ambient observations. Comparison of this technique to other fast GC techniques shows differences in the limit of detection by orders of magnitude (page 17, line 15). As discussed in section 4.5, this technique is, therefore, inferior to others but could still be useful for identifying monoterpenes based on fragmentation. This should be the main part of the abstract and conclusions sections. This should be further discussed in the manuscript, especially since the authors attempt to publish in an atmospheric measurement technique journal.

**Explicit measurements of LOD for our setup using MXT-Volatiles GC column were carried out and value the LOD is given in the revised text. The LODs were determined for $\alpha$ -pinene and R-limonene**

from analysis of a calibration curve as three times the standard error of predicted intercept value divided by the slope of the calibration regression line. The LoD of fast GC with SIFT-MS is inferior about factor of 10 to fastGC PTR-MS. As it was demonstrated by analysis of coniferous samples, this can still be used for rapid analyses of stronger monoterpene sources. We have also proposed in conclusions that an additional solution for improving sensitivity is using multicolumn capable to achieve higher sample flow.

In order to obtain valuable information, the authors suggest that changing ionization in the SIFT-MS is recommended. This implies that in order to obtain valuable information relative to other techniques the GC-SIFT should run in both ionization modes. What would be the time needed to go through an H3O+ and a NO+ cycle? How much more is the time compared to other fast GC techniques that only run once and with better resolution (page 17, line 16)?

**We have added this information to the revised manuscript.**

Overall, I would recommend that the value of this work and the comparison of this technique to others should be further discussed and emphasized throughout the manuscript.

**We have improved the discussion of the actual strengths and weaknesses in the conclusion.**

Section 2 is hard to read and I would suggest restructuring. In the first sentence of the section the authors introduce Fig. 1 but this is not followed by a discussion of the figure, the instrument parts, and operation. On the contrary, they discuss the column options and operating details and then go through the temperature profiles. I would recommend the following structure: A. A discussion of the parts of the fast GC preseparation system and the modes of operation with their details that are discussed in section 2.1 and page 4, line 15 to page 5, line 4, B. Operating details together with columns of choice and temperature profiles.

**We agree. We have changed the order so the article is more readable.**

In section 3.1 a short discussion regarding the humidity dependences is presented that is not supported by any figure or graph. Was the humidity of the different samples measured? If so, shouldn't these values be provided in all figures, especially since the effects seem to be substantial? Furthermore, this paragraph and further discussion should be part of the results and discussions and not the section it is now.

Effect of humidity on ion chemistry is substantial, however water has much shorter elution time and thus reaches SIFT-MS before the monoterpenes. If the temperature of the column is not very high and water CG peak do not overlap with the monoterpene peaks, water influence on ion product ratio will be small. The edge of the water peak can be, however, still influential for fastest monotepenes as apinene or camphene. The issue was clarified and more details about influence of water on ion chemistry are now discussed in the revised manuscript.

Section 4.1 and 4.2 have an overlap of results and discussion that makes these sections hard to follow. I would recommend that the authors work towards restructuring these sections to a clearer presentation of the results that the table and figures promote followed by a detailed discussion, for each graph, for each column, and the comparison of the two columns. A characteristic example of the difficulty of the reader to follow the results and discussion is the title of section 4.2 that has little to do with what is discussed in it. Furthermore, please discuss why NO+ was not tested for the MTXVolatiles column.

Sections 4.1 and 4.2 were carefully reconstructed to be more logical and easy to follow.

**Technical comments**

Title and manuscript: change "analyses" to "analysis"

**Corrected.**

Page 1, line 19: change to "...to separate them in less than 180 s...".

**Corrected.**

Page 2, line 15: change to ". . . which can affect human health. . . ".

**Corrected.**

Page 2, line 29: correct to "fast GC-PTR-ToF-MS" and in general correct throughout the manuscript "fastGC" to fast GC".

We are deliberately distinguishing between the general "fast CG" term describing the technique and "fastGC" witch is a trademark name of the commercially available module for PTR-MS, produced by Ionicon (https://www.ionicon.com/product/accessories/fastgc) and used in the referenced studies as factGC-PTR-ToF-MS.

Page 2, line 33: change to "we report method development results aimed to...".

Corrected.

Page 6, line 27: change to "are given in Table 1, and discussed in section 4".

Corrected.

Page 6, line 28: This is hard to follow sentence. Rephrase.

The sentence was reformulated.

Page 6, line 30: Change to "reagent"

Corrected.

Page 8, line 2: change to "saturation vapor pressures"

Corrected.

Page 17, line 27: change citation style

Corrected.

Page 17, line 17: Which results? What are the authors comparing here?

*We would like to compare limits of detection for SIFT-MS and PTR-MS setup. The sentence was clarified.*

Page 18, line 22-23: "... allows analysis of mixtures of monoterpenes in the air in short time periods. .." Is that the case for ambient measurements in the detection limits of the system? Isn't this overpromoting the capabilities of the system? For concentration above the detection limit (20 ppbv) is system able to analyse monoterpene mixtures. 60 s retention for MX-1 column as well as 180s for MXT-Vol column are considered as short time.

Table S1: It will be nice to add the m/z of detection.

We added the m/z information to the table.

The paper "Addition of a fast GC to SIFT-MS for analyses of individual monoterpenes in mixtures" by Lacko at all, submitted to AMT is work with potential. The addition of FastGC to the SIFT-MS is described in details, acknowledging most of the difficulties that method development brings. However, I find several major issues which prevent me from recommending its publication in AMT. First, the work is unnecessary long for the amount of information given. Two columns were compared, MXT-1 and MXT-Volatiles, but only the latter one gives acceptable separation. It is clear that MXT-1 is not suitable for this system (too fast separation, with not much control over the retention time). For any future version of the manuscript I suggest avoid the entire sections of MTX-1 column (perhaps it could be briefly mentioned in the supplementary data).

Second, the detection limit for this system is 100 ppb. Unfortunately, this is not close to the ambient levels of monoterpene concentrations or any plant chamber experiment loads. So, the relevance of this method is not within the scope of ATM, but rather in the fields where the technique can be used (monoterpene concentrations >100 ppb). Thus, I suggest to the authors to consider submitting these findings to a more suitable journal dealing with mass spectrometry techniques in general.

Minor comments: The manuscript in general needs more clarity: E.g. In the Abstract "the headspace of three conifer needle samples was analysed" it is not clear what do you mean here. The abstract should be clear and stand-alone. I believe you mean "needle samples of three conniver species"?

**Thank you for pointing this out. The statement was clarified.**

P4 L20. "(1 to 8 s)" is it 1 or 8 s you used? Or this is a range you can set? Again, I had to search in the following text to understand this better.

We clarified this information in the text so now it is stated what values were used in the measurements.

P8 L3-10. Not entirely clear enough. How did you enclose the plant branches? You mention temperature stress! But, how long it passed from the cutting? Did you use any light during the measurement?

We agree that this information can be important for description of a terpene emission. Plant branches were enclosed by wrapping the parafilm around the cut. Samples were measured app 30 minutes after harvest. Only a scattered light in the laboratory was presented. This information was added to the revised manuscript.

This work describes a GC-CIMS measurement technique developed to improve understanding of the composition of monoterpenes in the atmosphere which is an active area of interest in the atmospheric chemistry community due to key their roles in processes leading to formation of ozone and secondary organic aerosol (SOA) and is therefore highly relevant to the scope of AMT.

A series of experiments on individual standards of monoterpene isomers, monoterpene standard mixtures and the headspace of conifer foliage samples using a bespoke fast GC system coupled with a SIFT-MS is presented to demonstrate the potential application of fast GC-SIFT-MS for the separation and analysis of monoterpenes and other isomers in atmospheric and laboratory studies that is not currently achievable with SIFT-MS alone. The performance of two different GC columns in the fast GC SIFT-MS system was assessed - a generic (MXT-1) GC column and an application specific GC column (MXT-Volatiles). In addition, two reagent ions (NO+, H3O+) were used in the SIFT-MS system to aid in compound identification.

This work represents one of the first, if not the first, reported trial of a fast GC coupled with an SIFT-MS system which has a considerable user group worldwide. As noted in the manuscript introduction, this is an area of active development with previous papers describing fast GC coupled with other chemical ionization mass spectrometry (CIMS) systems, in particular PTR-MS (Materic et al 2015, Pallozzi et al 2016). Given the similarities between SIFT-MS and PTR-MS it could be considered that this paper does not represent a substantially novel development.

The original contributions to atmospheric measurement practice are:

1) The comparison of two GC columns - a generic (MXT-1) GC column (as used in previous fast-GC and GC-PTR-MS studies) and an application specific GC column (MXT-Volatiles) – this has relevance to the wider fast GC applications (SIFT-MS, PTRMS, other CIMS, fast GC-FID. . .) in which MXT-1 column has been used.

2) The first reported use of NO+ reagent ions in a fast GC - CIMS set-up.

However, additional additions/revisions are required for substantial conclusions to be reached regarding the performance and potential applications of fast-GC-SIFT-MS for quantification of monoterpene isomers. Specifically, more quantitative information is required on the detection limits, sensitivity and procedures for the quantification of species concentrations- see specific comments below.

**Specific comments**

Detection limit - p 17 Line 15 states "The present experiments indicate that using the fast GC-SIFT-MS combination, it is possible to achieve only qualitative analysis of the monoterpene mixture with a limit of the detection of about 100 ppb." Detection limits of 100 ppb is a major limitation for the application of fastGC-SIFTMS to measurements of individual monoterpenes in ambient air where concentrations are typically orders of magnitude lower (1 -10 ppb). The manuscript must include descriptions of:

1) How the stated detection limit of  $\sim$ 100 ppb was determined?

2) Why is this detection limit so high?

3) Potential improvements to the instrumental set-up that would reduce the detection limit to a range that would allow its application to measurements of ambient air (< 1 ppb).

All the raised questions are relevant are now discussed in the revised manuscript. The estimated LoD of 100 ppbv was not obtained by a proper calibration but just guessed. We have carried out additional determination of the LoD, using the MXT-Volatiles column. As the main limitation we identified the limited flow of the column witch decrease the sensitivity of the technique down to below 20 ppb. For higher temperatures the chromatograms have narrower peaks and LoD is much better (below 6 ppb). The limitation of SIFT-MS sensitivity depends on the total sample flow through the GC setup. This can be improved by using multiple parallel columns. All this discussion has been added to the revised manuscript.

Without these additions the application of this measurement technique for atmospheric measurements is limited making the relevance of this work to AMT highly questionable.

Quantification - The abstract, p 1 Line 18 states "Thus, it is possible to quantify components of a monoterpene mixture in less than 45 s by the MXT-1 column and to separate them in less 180 s by the MXT Volatiles column." Concentrations of monoterpenes are not quantified in this work and this claim is contradicted in the text p 17 Line 15 (as shown above) "it is possible to achieve only qualitative analysis of the monoterpene mixture". There are other similar contradictory statements in the manuscript which must be addressed.

This is again an important comment. Considering this and the comments of the other referee we have extended the study by including quantitative method of calculation and by obtaining calibration curves from which actual LODs were calculated and these are given in the revised manuscript.

Calibration – Wht is the sensitivity of this method? Was the system calibrated with certified gas standards containing one or more monoterpenes, and an empirical calibration factor determined?

We have carried out additional experiments to determine LoD and added that information to the revised manuscript. The sensitivity in the terms of retention times can be defined by the temperature profile, the length of the column and column type. The calibration was done using the diffusion tube method (Thompson and Perry, 2009) and the concentrations were determined by direct sampling SIFT-MS. This is now explained in Section 3.3

Absolute quantification - In lieu of an empirical calibration factor, the well-defined conditions in the SIFT-MS permit calculation of the concentrations of monoterpenes based on the raw signals of reagent and analyte ions (ie [m/z 137] as defined in section 3.2 of the manuscript), known reaction rates, and branching ratiosand instrument parameters as described in the SIFT-MS literature (e.g. Smith and Spanel 2005, Mass Spectrom. Reviews, 24, 661 – 700).

Yes, calculation of the concentrations u of monoterpenes sing the SIFT-MS is based on the raw signals of reagent and analyte ions. We have clarified this by adding a dedicated section dealing with quantification.

Direct measurement via SIFT-MS - Was direct quantification via SIFT-MS (without GC column) performed? Few comparisons of NO+ and H3O+ measurements of monoterpenes are available in the published literature and would be a valuable contribution.

Measurement of MS using the SIFT-MS was carried out and the results are now included in Supplement.

Both the detection limit and the sensitivity of the method are critical to understanding the application of this method for measurements of monoterpenes in the atmosphere and in laboratory studies. Neither are adequately described here making the relevance of this work to AMT highly questionable.

Relative abundance - In lieu of quantitative determination of individual monoterpene isomers, can the peak areas be used to estimate the relative abundance of each monoterpene species in the samples (mixtures and leaf headspace samples) ?

The peak areas be used, if separated, to estimate absolute concentration of each monoterpene. In not well separated chromatograms (as observed for MXT-1 column), absolute concentration cannot be properly estimated. However, using the additional regent ion, we can analyse coalesced peaks and determined in they do contain one or more monoterpenes. The discussion was improved to clarify these points.

Understanding the rel. abundance of monoterpenes is key to determining accurate calibration factors (see deGouw et al. (2003) JGR-Atmospheres 108, D21), and more importantly understanding the OH reactivity of BVOC dominated atmospheres. Suggest including NO+ and H3O+ reaction rates in Table 1 to demonstrate the importance of understanding the monoterpene composition to the accuracy of CIMS monoterpene measurements based on a single m/z, and adding a table of OH and O3 reaction rates for each monoterpene isomer identified and their relative abundance in leaf samples as well as some discussion regarding the potential contribution of different monoterpenes in the oxidation budgets of atmospheres dominated by emissions from these plant species. Overall, the measurement system and its operation are sufficiently explained however, inadequate information of the performance of this method in terms of detection limit and sensitivity are provided and potential future developments to improve performance are not adequately covered. Without this additional information the manuscript does not provide a substantial enough contribution to development of atmospheric measurement techniques for publication in AMT.

**We have added the required information and we changed the revised manuscript to clarify the importance of the isomeric analysis of monoterpenes.**

A key issue with CIMS instruments such as SIFT-MS and PTR-MS is essentially we know how much there is but we don't know what it is? Adding pre-separation techniques attempts to overcome this however, the data presented in this paper essentially reverses the challenge- we know what there is but not how much ? The manuscript requires a clear procedure for the quantification of monoterpene concentrations and/or the relative abundance of monoterpene isomers from the raw data in order to demonstrate the usefulness of this method over direct measurements with SIFT-MS. Quantification has been demonstrated in related instruments (Jones et al 2014, Materic et al 2015, Pallozzi et al 2016).and it is unclear why it was not part of this work.

**Quantification method and its results are now discussed in the revised manuscript.**

If these additions/revisions can be made, the following technical comments should also be considered.

**Technical comments**

Whole manuscript– replace SCI-MS with CIMS, the term chemical ionisation mass spectrometry (CIMS) is an established mass spectrometry term for analytical systems including SIFT-MS, PTR-MS etc

This project was funded from a EC project IMPACT involving 10 European institutions including those specialising in atmospheric research and the term Soft chemical-ionisation mass-spectrometry (SCIMS) is by consensus used to refer to SIFT-MS, PTR-MS and related techniques. Thus we prefer to keep SCIMS in this paper.

Abstract p1 line 18, change "quantify" to "qualitatively identify"

**Corrected.**

Abstract – add a couple of sentences at the end -what is the practical significance of this work? what is the theoretical significance?

P2 line 3, change "The analytical ion-molecule reactions" to "The chemical ionisation reactions"

**Corrected.**

P2, line 13, suggest addition of a new paragraph discussing the fact that due to issues with stability of monoterpene mixtures in certified gas standards, CIMS instruments employed in ambient air studies are often calibrated with certified gas standards containing only one or two monoterpenes, (typically a-pinene). However the instrument response differs between isomers due to differences in their ionization reaction rates and branching ratios. To determine an accurate (weighted) instrument sensitivity value for monoterpenes, the relative abundance of monoterpene isomers must be known (see deGouw et al. (2003) JGR-Atmospheres 108, D21).

A short paragraph mentioning the calibration issues with monoterpene standards was added to main text.

P2 paragraph lines 13 - 21 - these concepts need to be re-visited in discussion and summary to demonstrate the usefulness of these techniques.

**We have added the discussion to the conclusion.**

P2 line 21, move these two sentences into subsequent paragraph "Gas chromatography mass spectrometry (GC-MS) coupled with pre-concentration techniques has been developed to successfully identify and quantify different atmospheric monoterpenes (Janson, 1993; Räisänen et al., 2009; Song et al., 2015). However, the requirements of pre-concentration and long cycle time (more than 1h) are obviously unsuitable for real-time measurements."

**Both sentences were moved into a subsequent paragraph**

P4, "It is interesting to note that the flow of sampled air, established by the pressure difference between ambient atmosphere and the low pressure of the SIFT-MS flow tube, changes with the column temperature due to the variation of the dynamic viscosity of the air (see Fig. 2)." – Does this affect flow tube residence time (reaction time, t) important in SIFT-MS quantification calculations?

**It is affecting the total sample flow through the system and thus the calculation of quantifications. This effect can be estimated and included to the quantification calculation.**

P4, line 16, Can measurements by the SIFT-MS when the GC set-up is in "normal mode" be considered an instrument zero (SIFT-MS instrument background)? Can you use this data to calculate the detection limit and subtract from "sampling mode" measurements?

Yes, during the "normal mode" we can measure the background signal for selected masses, which can be considered as instrument zero. This value is usually found to be negligible; therefore, we don't

**have to subtract it. Information about detection limit obtained from calibration curves by the 3 sigma method was added to the revised manuscript.**

P5, line 16- "Sampling was repeated several times to improve sensitivity." No data for sensitivity is presented.

We agree, we changed the sentence.

P5 Section 3- insert details on the time it takes to switch between reagent ions and to achieve stable ion signals- this is crucial if NO+ and H3O+ are to be used for compound identification. What was the intensity and purity of the reagent ion signals?

We agree. Information was added to the revised manuscript. Switching between regent ions is very fast and require only tens of milliseconds. Purity of reagent ions is defend by the injection quadrupole, level of parasite ions is usually below 1%. Count rate of primary ions is usually in range of one million.

P7 insert section (after section 3.2) describing quantification procedure (as discussed in specific comments above) either using empirically derived calibration factors or via absolute quantification procedure based on [m/z 137] for H3O+ mode; and [m/z 136] for NO+ mode.

**We added a new section (Section 3.3) discussing the quantification.**

P8 Section 4.1 Comparison of columns: MXT-1 vs MXT-volatiles. The comparison of these two columns is valid given the use of the MXT-1 column in related instruments presented in the published literature (Jone et al 2014, Pallozzi et al 2016, Materic et al 2015 etc).

**We have discussed this in the revised manuscript.**

P8 paragraph line 12 -18 – Your approach needs to be more clearly articulated – for instance, firstly the instrument response to individual monoterpene species, in terms of retention time, and product ion ratios, was characterized via analysis of a series of prepared standards with both the MXT1 and MXT volatile columns and when H3O+ and NO+ were employed as the primary reagent ion in the SIFT-MS. Secondly, the separation of monoterpenes isomers using two columns, and the two reagent ions (NO+, H3O+) was demonstrated through analysis of prepared mixtures containing 8 monoterpenes. Lastly, the application of the GC-SIFT-MS for the separation (and quantification?) of monoterpene isomers in a real-world analysis is presented in a series of leaf headspace analyses.

**The initial paragraph was updated and clarified according to proposed schema.**

Section 3.3, Note it is unclear whether the same individual standards and mixtures of monoterpene were analysed by both NO+ and H3O+ in the same analysis runs?

**The same mixture was used and this is now stated in the revised manuscript.**

P8 line 22 – "Whilst the retention times for individual monoterpenes are different, they are not sufficiently stable (fluctuate by > 1 s, see Table 1) in the present fast GC device for analyses based on retention time only to be reliable." Suggested improvements to instrument design?

The fluctuation of retention may be caused by the fluctuation of the column temperature and therefore for longer column and lower temperature it may be reduced. (Effect will be less significant for longer retention times). This is now discussed in conclusion.

P8 line 28, the following statement is unclear "the peak shapes cannot be compared directly but the peak width (FWHM) increased only two times for the MXT-Volatiles column". Also define FWHM.

**The sentence was removed from the text.**

P9, Table 1 – add columns for reaction rates of monoterpenes with NO+ and H3O+ - consider landscape page layout (see comment above re Relative Abundance)

We have decided to keep the rate constants for the interaction of monoterpenes with regent ions in the Supplementary, together with the full list of potential products. The use of rate constant in the calculation of concentrations depends on all secondary ions produced from initial proton transfer, charge transfer of association interaction.

P11 Section 4.1 – Discussion of response to individual monoterpene standards. Insert Figure S2 and a corresponding plot for the MXT-volatiles column into section 4.1. These are very helpful when interpreting subsequent Figures 3 and 4. What conclusions can be reached from the tests of individual monoterpene standards – based on these tests what peaks are likely to co-elute, and what peaks are likely to be able to be separated in analysis of an unknown mixture? These tests provide the fundamental information for interpretation of the data from mixtures and leaf samples and should be included in the main text.

The plots of the response to individual monoterpene standards are quite busy and thus it is more clear to present them in the Table 1. Data are additionally directly shown in Fig 4 (bottom part) as horizontal lines, showing position of each monoterpene standard. For MXT-Vol column the identification is apparent. The discussion regarding identification of monoterpenes is clear in identification of monoterpene mixture, where for MXT-1: peak A is due to co-elution of  $\alpha$ -pinene, camphene and myrcene. Peak B is due to the presence of  $\beta$ -pinene exclusively and peaks C and D are due to the remaining four monoterpenes, mainly 3-carene and R-Limonen. We hope that the revised manuscript makes all of this clear.

P10 line 10 "As observed for both columns, separation can be improved by decreasing the column temperature (see Fig. S3 in the Supplement), however this may increase the chromatogram width and thus decrease the sensitivity of the technique. Additional sensitivity can be achieved by increasing the injection time, which will, however, increase the peak width." – this discussion is not quantitative, no explicit sensitivity data is presented.

**Unfortunately, we do not have quantitative data to demonstrate effect of injection time on sensitivity. The statement was corrected in revised manuscript.**

The discussion in Section 4.1 regarding analysis of mixtures needs to be restructured.:

1) provide a direct comparison between MXT-1 and MXT-volatiles at the same conditions. (~40- 45C). Figures 3 and 4 – Figure 3 is actually a comparison of H3O+ and NO+ and the data from the MXT1 and MXT-volatiles column are not compared side-byside. Format a page in landscape orientation, combine figures 3 and 4 (three panels) and present them in a compatible format (ie same formatting and labelling etc).

**Direct side-by-side comparison of both columns is now given in the revised manuscript at room temperature and at 40C.**

2)Discuss challenges and potential improvements ie stability in retention times, improved separation via decreasing column temp, improved sensitivity by increasing injection times.

Discussion regarding stability and separation is clarified in the revised manuscript.

3)Present MXT-volatiles column data under optimized conditions – ie "The MXTVolatiles column facilitates identification of all monoterpenes present in the mixture for temperatures close to room temperature (see Fig. S3 in the Supplement)." – the top panel in the S3 plot is key to demonstrating the achievable separation of the MXTvolatiles column - move it from the supplement to the main body. The additional species identifiable using this technique compared to the MXT-1 set-up need to be more clearly summarised.

**The room temperature data are compared side-by-side fir both columns. The discussion of the comparison of the two columns is improved by reorganisation of the sub sections.**

P12 Paragraph lines 8 – 17- needs to be moved to later in the discussion or into section 4.5 to show that aside from potentially better selectivity other co-benefits of employing the NO+ reagent ion in CIMS measurements of BVOCs, in particular in measurements of isoprene (See Karl et al 2012 ACP 12:11877 – 11884, and Karl et al 2014 Int J. Mass Spectrom. 365-366:15-19). There are many more species which interfere with quantification of isoprene in H3O+ reagent ion mode such as furan, 2,3,2-MBO, C5 aldehydes.

**Discussion regarding benefits of NO+ reagent ion was extended to discuss potential benefit of identification of isoprene.**

P12 line 19 " However, the ratios obtained for  $\alpha$ -pinene and myrcene are somewhat variable between the FS and MIM data and they also differ somewhat from the literature values." – be quantitative ie state % variability. Is the variability a result of changes in the reagent ion intensity (consider using normalised intensity), or composition (eg % reagent ion impurities of H3O+(H2O), O2+, NO+)?

The ion ratio is not dependent of the ion intensity or ion impurities, however, it can be affected by secondary processes with neutral water molecules or hydronium hydrates. The issues of water influence were clarified in the text. Changes of the ion ratio are now discussed in the revised manuscript.

P14 Section 4.3 –For this method to be useful in atmospheric research the concentrations of monoterpene isomers or an estimate of their relative abundance must be quantified from the data and presented here and section 4.4(see specific comments above re quantification).

**Re-analysis of data was carried out to obtain concentrations of detected monoterpenes and thus demonstrate usefulness of the technique in atmospheric research.**

P14 Section 4.4- be consistent – use dot point format as for previous section. Why is the data from non-optimized conditions (40C) presented? Was the analysis done at the optimal temperature (5V) for separation? If so, should be presented.

Dot point format is now used for this section in the revised manuscript. Analysis was carried out using 40C conditions only as we would like to compare MXT-1 and MXT-Vol columns at same conditions. Conditions optimal for separation are not applicable for SCI-MS techniques, as the separation need 700s to be provided. At temperature 40C is sufficient separations provided under 180 s.

P14 line 14, " The signal increase in the third region may indicates trace presence of (R)-(+)-limonene." – the m/z81 signal or the ion intensity?- not clear.

P15 Section 4.4- need to state that similar experiments but on a different series of conifer samples were also conducted using the MXT-volatiles column.

**The information was added in the revised manuscript.**

P15 Figure 5- consistent units (normalised intensity) should be used for all figures (3-6), label peaks in both H3O+ and NO+ chromatograms (both Fig 5&6). Query the signal to noise ratio of some of the identified peaks e.g. H3O+ spruce 3-carene / limonene peak. Re-iterates importance of quantifying method LoD.

Labelling in figures was modified. The observed fluctuation/variation in Fig 5-6 is caused by real signal representing presence of monoterpenes. The background intensity is close to 0. The signal to noise ratio is more than 300.

P17 Section 4.4 – This section should conclude with a table of the relative abundance of each monoterpene isomer in the leaf samples and their reaction rates with OH and O3 with associated discussion.

The calculated absolute concentrations of detected monoterpenes were added to the text. Table shoving the OH and O3 reaction rates with monoterpenes was created and include in the introduction. Discussion regarding importance of monoterpene separation and their different OH and O3 reactivity was added to conclusions.

P17 Section 4.5 – "The present experiments indicate that using the fast GC-SIFT-MS combination, it is possible to achieve only qualitative analysis of the monoterpene mixture with a limit of the detection of about 100 ppb. This is inferior to the previously described fastGC-PTR-MS systems (Materic et al., 2015; Pallozzi et al., 2016), which ' achieved full separation with limit of the detection up to 1-2 ppt." – list the reasons for the difference in performance and potential future developments of the GC-SIFT-MS method to improve performance. This statement must be addressed in more detail as these significant limitations preclude the application of this method to ambient studies and make the inclusion of this work in AMT questionable.

**We have carried out new experiments and obtained LOD from calibration curves. Ppt was a mistake, in Materic et al., 2015; Pallozzi et al., 2016 1-2 ppb was achieved.**

P17 line 17 – start new paragraph at "However, one advantage of SIFT-MS is the facility to use two reagent ions, and the analysis of product ion ratios provides additional information. Thus, the combination of the data from the two reagent ions together with the analyses of the product ion signal ratios ri can be shown to improve the identification of monoterpenes." – be more specific, what additional compounds were identified using the reagent ion chemistry. Suggest insert discussion from 4.2, on usefulness of NO+ reagent ion for identification of other BVOCs here. As a side note, switchable reagent capability has been developed for PTR-MS and other CIMS and is not unique to SIFTMS.

**Myrcene and camphene were stated as examples of monoterpenes that benefit from use of NO+ regent ion. The discussion was moved from section 4.2.**

P17 line 20 – "The results obtained from the present study agree well with the literature reports." Be more specific, suggest – the results obtained from the analysis of leaf headspace samples agree well other studies in the published literature. Suggest authors present comparisons by tree species as a table with following columns plant species name; monoterpenes identified; rel. abundance where available; measurement method; time resolution; and where available: LoD & sensitivity; and literature reference. Focus discussion on number and rel. abundance of monoterpenes identified and the methods used, not on geographical variability or variability between species beyond the scope of this work. What is the potential advantage of this method over others? Time resolution?

The detail analysis and detail comparison between plants and their monoterpene concentration is behind the scope of this publication. We do not focus on geographical origin of samples. It is well known that concentration and composition of monoterpenes is very sensitive to the sampling technique. We have edited this section to discuss percentages rather than absolute concentrations.

P18 Section 5- "A new method has been developed that allows quantitative analyses of individual monoterpenes in mixtures using SIFT-MS enhanced by chromatographic preseparation." As previously stated this is not correct and contradicts the first line of the previous section (4.5) "The present experiments indicate that using the fast GC-SIFTMS combination, it is possible to achieve only qualitative analysis of the monoterpene mixture with a limit of the detection of about 100 ppb."

P18 line 16 start new paragraph at "A weakness of the current fast GC setup is the relatively poor temperature stability caused by a strong dependence on the laboratory ambient temperature. . .."

**Corrected.**

P18 line 18 "It has been shown that a clear advantage of SIFT-MS is the facility to use different reagent ions and to utilize the ratios of the specific product ions of their reactions with the various monoterpene isomers at the same retention time to improve the identification of the monoterpenes." Belongs in previous paragraph (P18, line 10).

**We have rearranged the order.**

P18 line 23 – "This novel idea of a fast GC-SIFT-MS combination could broaden the application of SIFT-MS to in situ trace gas analyses of complex mixtures such as ambient air and exhaled breath.". There are several issues with this statement: 1) SIFT-MS is already used for in situ ambient air and breath analysis- this technique GC-SIFTMS does not broaden its application. The practical significance of this work is that it aims to address the challenge of quantifying isomers in CIMS measurements of complex mixtures. 2) Also, need to preface this statement "With improved limits of detection and sensitivity, this novel fastGC-SIFT-MS could. . . . . ..." currently its application in ambient air analysis is limited due to high LoD and lack of data about its sensitivity.

**We have rewritten the conclusion with these points in mind so it truly represents the outcome of this work including the additional LOD determinations.**

What is the theoretical significance of this work- what will an improved understanding of the complex mixture of monoterpenes contribute to our understanding of atmospheric chemistry? Ie estimates of total OH reactivity etc.

The detailed answer to this is outside of the scope of this paper, nevertheless we hope that the revised manuscript indicates the further direction in fast GC SIFT-MS development.

**Addition of a fast GC to SIFT-MS for analyses analysis of individual monoterpenes in mixtures**

Michal Lacko1,2, Nijing Wang3, Kristýna Sovová1, Pavel Pásztor1, Patrik Španěl1

1The Czech Academy of Science, J. Heyrovský Institute of Physical Chemistry, Dolejškova 2155/3, 182 23 Prague, Czech
5 Republic

2Faculty of Mathematics and Physics, Charles University in Prague, Ke Karlovu 3, 121 16 Prague, Czech Republic 3Air Chemistry Department, Max-Planck-Institut für Chemie, Hahn-Meitner-Weg 1, 55128 Mainz, Germany

Correspondence to: Michal Lacko (michal.lacko@jh-inst.cas.cz)

[revised manuscript text omitted]

a taken from Atkinson (Atkinson and Arey, 2003) unless noted otherwise.

b Assumed OH radical concentration: 2.0x106 molecule cm-3, 12-h daytime average.

c Assumed O3 concentration: 7x1011 molecule cm-3, 24-h average.
 d Lifetimes are estimated in relation to [NO3] = 10 ppt, [O3] = 20 ppb for night; and [OH] = 106 molecules per cm3, [O3] = 20 ppb for day light conditions. (Kesselmeier and Staudt, 1999) (unless noted otherwise)
 e Rate constants (in units of 10-17 cm3 molecule-1 s-1) for the gas-phase reactions of O3 with a monoterpenes have been determined at 296 ± 2 K and 740 torr total pressure of air or O2 using a combination of absolute and relative rate techniques.
 (Atkinson et al., 1990) (unless noted otherwise)

f Rate constants (in units of 10-11 cm3 molecule-1 sec-1) for the gas-phase reactions of the OH radical with monoterpenes have been determined in one atmosphere of air at 294 ± 1 K. (Atkinson et al., 1986) (unless noted otherwise)
 g Rate constants of k(OH + isoprene) = 1.01 × 10-10 cm3 molecule-1 s-1. O3 reaction rate constants determined in 10-19 cm3 molecule-1 s-1 units. OH radical reaction rate constants determined in 10-11 cm3 molecule-1 s-1 units. (Atkinson et al., 1990)

15 nd – no data

Gas chromatography mass spectrometry (GC-MS) coupled with pre-concentration techniques has been developed to successfully identify and quantify different atmospheric monoterpenes (Janson, 1993; Räisänen et al., 2009; Song et al., 2015). However, the requirements of pre-concentration and long cycle time (more than 1h) are obviously unsuitable for real-time

20 measurements.

A promising approach to the near real time analysis of isomeric molecules is to combine both SCI-MS and fast GC methods. Pre-separation provided by fast GC involves short columns with thin active layers, fast temperature ramps, fast injection systems and time resolutions below 5 min (Matisová and Dömötörová, 2003). Materic et al. (Materić et al., 2015) established a system using PTR-MS coupled with a fast GC to detect individual monoterpenes and achieved the separation of six most

25 common monoterpenes at a limit of detection down to 1.2 ppbv. Pallozzi et al. then compared a fastCG-PTR-ToF-MS system

with traditional GC-MS methods, discussing the limitations of the fast GC setup on some BVOCs emitted from plants, including monoterpenes (Pallozzi et al., 2016). SIFT-MS is also widely used in VOCs analyses (Allardyce et al., 2006; Smith and Spaněl, 2011b, 2005b). It has well-defined analytical reaction conditions and the  $H_3O^+$ ,  $NO^+$  and  $O_2^{+\bullet}$  reagent ions can be switched rapidly to analyse time-varying trace gases in air samples. In the present article, we report the results of method development results aimed atto selective analyses of individual monoterpenes in mixtures in air using a bespoke fast GC/SIFT-MS combination with  $H_3O^+$  and  $NO^+$  reagent ions. This involved the analysis of both prepared laboratory monoterpene/air

5

mixtures and headspace of the foliage of different pine trees.

Figure 1: Schematic visualization of the fast GC-SIFT-MS experiment, Coloured dashed lines in the inlet part of the fastCG fast CG 10 represent gas flow through the system of the valves EV1-3. The blue line traces the "normal mode" regime, the green line represents the "sampling mode" and the red line represents the "cleaning mode".

**2 Construction of a fast GC device for pre-separation**

The experimental setup of the bespoke fast GC setup constructed as an addition to SIFT-MS is shown in Fig. 1. In the two different GC columns were tested. First, a 5 m long nonpolar general purpose chromatography metallic experiments.

15 0.1 um active phase, Restek Inc.) using dry air as the carrier gas, which was chosen according DTD MS factGC analyses 2014) Additionally a second application specific column for volatile organic pollutants, MXT-Volatiles (0.28 mm × 1.25 um active phase, Restek Inc.), was used with helium carrier gas. In order to facilitate direct resistive heating, the coil-shaped stainless steel columns (resistivity ~1.2 Ω/m) were electrically isolated and connected to a regulated 60 V, 5 A DC power supply. Appearance of cold spots was suppressed by ensuring that the electrical eurrent runs through the entire length of the columns. The temperatures of the columns were monitored by a K-type probe connected to their centres (see the right part of Figure 2 for the temperature variation with applied voltage). It is interesting to note that the flow of sampled air established by the pressure difference between ambient atmosphere and the low pressure of the SIFT-MS flow tube changes with the column temperature due to the variation of the dynamic viscosity of the air (see Fig. 2).

---

## Referee Report (RR1)

**Referee report on* Addition of fast GC to SIFT-MS for separation and analysis of monoterpene isomers**

Michal Lacko1,2, Nijing Wang3, Kristýna Sovová1, Pavel Pásztor1, Patrik Španěl1

1The Czech Academy of Science, J. Heyrovský Institute of Physical Chemistry, Dolejškova 2155/3, 182 23 Prague, Czech Republic 5
2Faculty of Mathematics and Physics, Charles University in Prague, Ke Karlovu 3, 121 16 Prague, Czech Republic
3Air Chemistry Department, Max-Planck-Institut für Chemie, Hahn-Meitner-Weg 1, 55128 Mainz, Germany

**General Comments**

This work describes a GC-CIMS measurement technique developed to improve understanding of the composition of monoterpenes in the atmosphere which is an active area of interest in the atmospheric chemistry community due to key their roles in processes leading to formation of ozone and secondary organic aerosol (SOA) and is therefore highly relevant to the scope of AMT.

A series of experiments on individual standards of monoterpene isomers, monoterpene standard mixtures and the headspace of conifer foliage samples using a bespoke fast GC system coupled with a SIFT-MS is presented to demonstrate the potential application of fast GC-SIFT-MS for the separation and analysis of monoterpenes and other isomers in atmospheric and laboratory studies that is not currently achievable with SIFT-MS alone. The performance of two different GC columns in the fast GC SIFT-MS system was assessed - a generic (MXT-1) GC column and an application specific GC column (MXT-Volatiles).In addition, two reagent ions ($NO^+$, $H_3O^+$) were used in the SIFT-MS system to aid in compound identification. The methods and assumptions described in the manuscript are valid, clearly outlined and reproducible.

This work represents one of the first, if not the first, reported trial of a fast GC coupled with an SIFT-MS system which has a considerable user group worldwide. As noted in the manuscript introduction, this is an area of active development with previous work describing fast GC coupled with other chemical ionization mass spectrometry (CIMS) systems, in particular PTR-MS (Materic et al 2015, Pallozzi et al 2016) are properly acknowledged in the manuscript.

Given the similarities between SIFT-MS and PTR-MS it could be considered that this paper does not represent a substantially novel development.

The original contributions to atmospheric measurement practice are:

- The comparison of two GC columns - a generic (MXT-1) GC column (as used in previous fast-GC and GC-PTR-MS studies) and an application specific GC column (MXT-Volatiles) – this has relevance to the wider fast GC applications (SIFT-MS, PTR-MS, other CIMS, fast GC-FID...) in which MXT-1 column has been used.

- The first reported use of $NO^+$ reagent ions in a fast GC - CIMS set-up.

The presentation is generally well structured however the language is at times unclear and the manuscript requires significant copy editing prior to final submission. Furthermore, specific comments below should also be addressed.

**Specific Comments**

- **P1 lines 24 - 26** – suggest re-write "The system can thus be used for direct rapid monitoring of monoterpenes above 20 ppbv, such as applications in laboratory studies of monoterpene standards and leaf headspace analysis. Limitation of the sensitivity due to the total sample flow can be improved using a multicolumn pre-separation.".

- **P2 lines 12 – 14** – " However, chemically similar molecules with the same atomic composition (structural isomers) usually produce identical analyte ions with similar branching ratios and therefore the neutral analyte molecules cannot be easily differentiated using SCI-MS alone (Smith et al., 2012)". *Suggest addition* - As a result, standard SCI-MS techniques such as SIFT-MS and PTR-MS are limited to reporting concentrations of the sum of monoterpenes present in the sample, and the composition of monoterpenes present cannot be determined.

- **P2 lines 19 – 27** – move paragraph to start of Introduction, then follow on with introductory discussion of measurement of monoterpenes.

- **P4 line 1** – what were the findings of the discussion in Pallozi et al relevant to this section? The relevant limitations?

- **P4 lines 4 – 7** – suggest re-write "In the present article, we report method development results aimed to  selectively analyse individual monoterpenes in mixtures in air using a bespoke fast GC/SIFT-MS combination with H3O+ and NO+ reagent ions.."

- **P6 line 11** – "This effect can to be estimated and have to be included to a quantification calculation." *Re-write* – This effect can be estimated and has to be included in the quantification calculation". *Further detail needed for reproducibility of results– the effect is estimated how? And how is the estimation included in the quantification calculation described in section 3.3?*

- **Section 3.1** – Why is exothermicity relevant? – The stability of the product ion molecules depends on their internal energy which in turn depends on the exothermicity of the chemical ionisation reactions. The exothermicity of NO+ reactions is discussed, but not the exothermicity of PTR from H3O+. The exothermicity of proton transfer from H3O+ is given by the proton affinity (PA) of the H2O (165 kcal mol-1) minus the PA of the analyte.

  "The interaction of the primary ions with monoterpenes may be affected by presence of neutral water molecules and thus by different humidity of the sample". In the case of H3O+ this is due to the presence of H3O+(H2O)n reagent ion clusters which also undergo PTR with monoterpenes and have a higher PA than that of H2O (PA (H2O.H2O) = 195 kcal mol-1) and the PTR reaction is less exothermic thus reducing fragmentation.

- **Section 3.3** – add detail of temperature effect mentioned previously on P6 (see above).

- **P10 lines 12 – 15** – In the present experiment  both columns were heated isothermally to the temperature app. 40 □C  selected to optimise both temperature stability and chromatographic separation (see Fig S4 in supplement). For higher temperatures, the monoterpene chromatogram peaks coalesced while for lower temperatures a significant influence of the lab air temperature fluctuations was apparent. However, even At these

optimised conditions for MXT-1 column, monoterpenes are not fully separated and thus, fast GC with MXT-1 column alone (at 40 □C) provides only qualitative analysis.

- **P16 lines 13 – 16** – clarify "The second region of a small peak 0.38 (H3O+) and 0.14 (NO+)."

- **Section 4.4** – As we do not know the composition of the monoterpenes in the leaf samples and are comparing to only 8 standards, consider less definitive language in places e.g. for Spruce: "The tailing edge of the first peak shows a decrease of $r\overline{w}$ (0.29 for H3O+, 0.14 for NO+)  attributed to a small contribution by camphene.

- **P22 lines 4 – 9** – The description of LOD determination and results should come earlier - move to section 3.3 re-name the section "Fast GC SIFT-MS limits of detection and quantification". Only comparison with LOD of other similar techniques (e.g. GC-PTR-MS )- why are they different?, as well as proposed developments to improve LOD of this system should be discussed here in section 4.5.

- **P22 lines 21 – 24** - *re-write* "Aside from potentially better selectivity, a benefit of employing the NO+ regent ions in atmospheric analysis is quantification of isoprene, which for $H_3O$+ reagent ion mode, suffers mass interference from product ions of other biogenic species including  furan, C5 aldehydes and 2-methyl-3-buten-2-ol (Karl et al., 2012; Karl et al., 2014 as well as the second hydrate of methanol that is also emitted by plants (12% of global BVOC emissions) (Španěl et al., 1999).

- **P23 line 28** – *suggest adding-* The major limitation of the GC-SIFT-MS system described here is the high limits of detection (~16 ppb) which currently preclude its application in measurements of monoterpenes in typical ambient concentrations.

- **P23 lines 26 – 27** – *This statement is unclear, consider re-writing* "The fast GC SIFT-MS combination can thus be a step towards atmospheric analyses of monoterpenes that should resolve individual compounds due to their different reactivity with the OH radicals." – due to their different OH reactivities, the ability to distinguish individual monoterpenes at high time resolution with fast GC SIFT-MS has the potential to improve our understanding of the contribution of individual monoterpenes in atmospheric chemistry processes such as formation of tropospheric ozone and SOA.

---

## Editor Decision (ED1)

Page and line numbers are for the main text of author_response_version4.pdf

Page 2 Table 1 and elsewhere: Change "R-limonene" to "(R)- limonene".

Page 4 Figure 1, page 12 Table 2 footnote, page 17 line 5, page 23 line 18 and page 25 line 11: Change "GC-SIFT-MS" to "GC/SIFT-MS".

Page 5 line 10: Change "°C.min$^{-1}$" to "°C min$^{-1}$".

Page 7 lines 12 and 14: Change "°um" to "µm".

Page 8 line 18: Change "$\Omega$/m" to "$\Omega$ m$^{-1}$".

Page 8 line 29: Change "cm.s$^{-1}$" to "cm s$^{-1}$".

Page 9 line 8: Change "a-pinene" to "α-pinene".

Page 9 lines 8 and 9: Change "KJ.mol$^{-1}$" to "KJ mol$^{-1}$".

Page 11 line 8: Change "L/mol" to "L mol$^{-1}$".

Page 11 line 14: Change "cm$^3$s$^{-1}$" to "cm$^3$ s$^{-1}$".

Page 11 line 14: Change "c/s" to "c s$^{-1}$".

Page 11 line 18: Change "°um" to "µm".

Page 11 line 29: Change "ml" to "mL".

Page 17 lines 24 and 25: Remove a space between a number and a plus minus sign.

Page 17 line 31: Add "OriginLab Corporation, Northampton, MA, USA" for the citation of Origin 9.0.

Line 19 lines 17-32: Remove a space between a number and a plus minus sign.

---

## Author Response (AR2)

**Authors response**

**Referee 1**

Thank you for the list of specific comments. The document with the comments is unfortunately not editable, therefore we cannot answer to the comments side by side. However, all your valuable comments are very pertinent and we have modified the manuscript accordingly. Many thanks. The manuscript is now greatly improved.

Pallozi et all discuss the limitation of the short 6m GC column. We found that some isobaric molecules (e.g. b-pinene and sabinene) emitted from plants were not properly separated in the fastGC PTR-ToF-MS setup and they recommended using longer columns and fast temperature gradients to improve the separation. This is now clear in the revised manuscript.

The effect of variation of a dynamic viscosity of gas (air) causes a change of the column flow rate at different column temperature. That affects the total amount of sample carried onto the analytical instrument. Therefore, the column flow has to be measured for any temperature used. The value of column flow is then used in Eq. 8 to estimate the sampled volume V. Hopefully, this is now clear in the reviewed manuscript.

The exothermicity of the ion-molecule reaction process is relevant as it defines if the processes of charge transfer or proton transfer can effectively occur. The exothermicity for  $NO^+$  regent ion was mentioned, because  $NO^+$  has lower ionization energy than the  $O_2^+$  regent ion.  $NO^+$  often form adducts with neutral organic species, but with monoterpenes reacts via charge transfer and dissociative charge transfer as the difference in ionization energy is more than 1 eV. The proton affinity is known only for three monoterpenes. Based on the PA we may show that even higher hydronium clusters are able to react with monoterpenes via proton transfer and thus reduce the fragmentation as the exothermicity of proton transfer for higher water clusters is reduced. The values of PA are now discussed in the reviewed version of the manuscript.

**The problem at P16 lines 13-16 was clarified.**

The discussion focusing the LOD was placed into the section 3.3 and difference between PTR and SIFT in terms of LOD are now discussed in section 4.5 in the revised manuscript together with proposals for the development to improve the LOD.

**Referee 3**

The authors did a decent job trying to incorporate the suggested comments from the reviewers. However, the English in the manuscript should be improved and I strongly recommend a language service to support the manuscript before final publication. Certain minor comments should also be addressed regarding the structure of the paper. A few examples of English mistakes are captured in the technical notes section below.

Thank you for these valuable comments. All of them have been acted upon and the manuscript is greatly improved as a result. The quality of English in the revised manuscript has been improved by a native English speaker.

**Specific comments**

In section 4.4 the authors suggest that for the tree samples only MTX-Volatiles column was used but in the same section a discussion on the comparison of the two columns is performed that is confusing. Also, I consider that this section is lacking further discussion by the authors regarding its comparison to the laboratory results and the uncertainties. For example, the complexity of the tree samples with hexenal should be part of this section and not the section "comparison to other studies".

The discussion in the section 4.4 has been improved and information about uncertainties were added. Additionally, we moved all the discussion regarding the comparison of the two columns and interference with 3- hexanal into the new section 4.5.

Discussion of the LOD and calibrations is currently presented in section 4.5. I would consider this information to be a section of its own at the beginning of the manuscript and not the end. Still, the comparison of the LOD and calibrations to other work should stay at the comparison section as is.

**The discussion of the LOD is now in Section 3.3 while the comparison with other work remained in the Section 4.5.**

**Technical comments**

Page 7, line 5: change to "Typical count rate of the reagent ions...".

Page 7, line 30: change to "... by the presence..."

- Page 7, line 32: change to "... that resulted in the decrease of the product..."
- Page 7, line 33: change to "... reagent..." and correct throughout the manuscript

Page 8, line 1: change "or" to "and"

Page 9, line 12: change "by" to "following"

- Page 9, line 13: change "was" to "were"
- Page 10, line 16: change "Table 2" to "Table 2, and further supported by Figure S2"
- Page 14, line 18: change "he is..."
- Page 14, line 18: change to "...GC..."
- Page 14, line 23: change to "... and 0.40 respectively..."
- Page 14, line 23: add standard deviation of values as 2 XX

Page 14, line 25: rephrase sentence "Do dynamic..."

- Page 15, line 31: change to "according to the expected..."
- Page 15, line 32: change to "... compared..."
- Page 22, line 32: change to "... compared..."
- Page 23, line 18: this is not quantitative analysis when a separation is not possible. Please rephrase.

**Addition of a fast GC to SIFT-MS for analysis of individual monoterpenes in mixtures**

Michal Lacko1,2, Nijing Wang3, Kristýna Sovová1, Pavel Pásztor1, Patrik Španěl1

1The Czech Academy of Science, J. Heyrovský Institute of Physical Chemistry, Dolejškova 2155/3, 182 23 Prague, Czech 5 Republic

2Faculty of Mathematics and Physics, Charles University in Prague, Ke Karlovu 3, 121 16 Prague, Czech Republic 3Air Chemistry Department, Max-Planck-Institut für Chemie, Hahn-Meitner-Weg 1, 55128 Mainz, Germany

Correspondence to: Michal Lacko (michal.lacko@jh-inst.cas.cz)

- Abstract. Soft chemical ionization mass spectrometry (SCI-MS) techniques can be used to accurately quantify volatile
  organic compounds (VOCs) in air in real time; however, differentiation of isomers still represents a challenge. A suitable pre-separation technique is thus needed, ideally capable of analyses in a few tens of seconds. To this end, a bespoke fast GC with an electrically heated 5 m long metallic capillary column was coupled to selected ion flow tube mass spectrometry (SIFT-MS). To assess the performance of this combination, a case study of monoterpene isomer (C10H16) analyses was carried out. The monoterpenes were quantified by SIFT-MS using H3O+ reagent ions (analyte ions C10H17+, *m/z* 137, and
- 15 C6H9+, m/z 81) and NO+ reagent ions (analyte ions C10H16+, m/z 136, and C7H9+, m/z 93). The combinations of the fragment ion relative intensities obtained using H3O+ and NO+ were shown to be characteristic offer the individual monoterpenes. Two non-polar GC columns (Restek Inc.) were tested: the advantage of MXT-1 was shorter retention whilst the advantage of MXT-Volatiles was better separation. Thusa it is possible to identify components of a monoterpene mixture in less than 45 s by the MXT-1 column and to separate them in less than 180 s by the MXT-Volatiles column. Quality of separation and
- 20 sensitivity of present technique (LOD ~16 ppbv) was found to be inferior compared to commercially available fast-GC solutions coupled with proton transfer reaction mass spectrometry (PTR-MS, LOD ~1 ppbv) due to the limited sample flow through the column. However, using combinations of two reagent ions improved identification of monoterpenes not well resolved by in the chromatograms. As an illustrative example, the headspace of needle samples of three conifer species was analysed by both reagent ions and with both columns showing that mainly α-pinene, β-pinene and 3-carene were present.
- The system can thus be used for direct rapid monitoring of monoterpenes above 20 ppbv, such as applications in laboratory studies of monoterpene standards and leaf headspace analysis.- Limitation of the sensitivity due to the total sample flow can be improved using a multicolumn pre-separation.

**1** Introduction**

Monoterpenes, mostly emitted from plants, are very important biogenic volatile organic compounds (BVOCs) in the atmosphere. Due to their high reactivity with atmospheric oxidants such hydroxyl radicals (OH\*), oxidation of monoterpenes can lead to tropospheric ozone ( $O_3$ ) accumulation as well as to secondary organic aerosol formation, which can affect human health and contribute to global climate change (Chameides et al. (1992); Fehsenfeld et al. (1992); Kulmala et al. (2004)). Although all monoterpenes comprise two isoprene units and all have the same molecular formula,  $C_{10}H_{16}$ , their lifetime (inverse to reactivity) for reaction with OH• and  $O_3$  widely varies from minutes to days (Atkinson and Arey, 2003) (See Table 1). The values of the total OH reactivity, which is dominated by BVOCs measured in rainforests, have been found to be higher than expected, which could be attributed to undetected monoterpenes or sesquiterpenes (Nolscher et al., 2016). Therefore, it is important to identify and individually quantify these BVOCs at their ambient trace levels.

| Compound    | Lifetime for reaction with a      | Chemical          | lifetimed | Rate constant of O 3 | Rate constant of OH         |
|--------------------|-----------------------------------|-------------------|-----------------------------|---------------------------------|-----------------------------|
|                    | $OH^{\rm b}$                      | Day               | Night_                      | reaction e           | reactionf |
|                    | $\underline{O_3}^{\underline{c}}$ |                   |                             |                                 |                             |
| a-pinene    | 2.6 hrs                    | 2-3 hrs    | 5-30 min             | 8.7                      | $5.45 \pm 0.32$             |
|                    | 4.6 hrs                    |                   |                             |                                 |                             |
| β-pinene    | 1.8 hrs                    | 2-3 hrs    | 5-30 min             | 1.5                      | $7.95 \pm 0.52$             |
|                    | 1.1 day                    |                   |                             |                                 |                             |
| camphene    | 2.6 hrs                    | nd         | nd                   | 9.0g          | 5.33g     |
|                    | 18 day                     |                   |                             |                                 |                             |
| myrcene            | 39 min                     | 40-80 min  | 5-20 min             | 49                       | $21.3 \pm 1.6$              |
|                    | 50 min                     |                   |                             |                                 |                             |
| 3-carene    | 1.6 hrs                    | nd                | nd                   | 3.8                      | $8.70 \pm 0.43$             |
|                    | 11 hrs                     |                   |                             |                                 |                             |
| R-limonene  | 49 min                     | 40-80 min  | 5-20 min             | 21                       | $16.9 \pm 0.5$              |
|                    | 2.0 hrs                    |                   |                             |                                 |                             |
| a-terpinene | 23 min                     | < 5 min | $\leq 2 \min$               | 870                      | $36.0 \pm 4.0$              |
|                    | $\frac{1 \min}{1}$                |                   |                             |                                 |                             |
| y-terpinene | 47 min                     | nd         | nd                   | 14                       | $17.6 \pm 1.8$              |
|                    | 2.8 hrs                    |                   |                             |                                 |                             |

**Table 1. Monoterpenes included in the present study listed together with their atmospheric lifetime.**

5

a taken from Atkinson (Atkinson and Arey, 2003) unless noted otherwise.
 b Assumed OH radical concentration: 2.0x106 molecule cm-3, 12-h daytime average.
 c Assumed O3 concentration: 7x1011 molecule cm-3, 24-h average.
 d Lifetimes are estimated in relation to [NO3] = 10 ppty, [O3] = 20 ppb for night; and [OH] = 106 molecules per cm3, [O3] = 20 ppb for day light conditions. (Kesselmeier and Staukt, 1999) (unless noted otherwise)

a Rate constants (in units of  $10^{-17}$  cm3 molecule-1 s-1) for the gas-phase reactions of O3 with a monoterpenes have been determined at 296 ± 2 K and 740 Torr total pressure of air or O2 using a combination of absolute and relative rate techniques. (Atkinson et al., 1990) (unless noted otherwise)

 $\frac{f}{Rate constants (in units of 10^{-11} cm^3 molecule^{-1} sec^{-1}) for the gas- phase reactions of the OH radical with monoterpenes}{have been determined in one atmosphere of air at 294 ± 1 K. (Atkinson et al., 1986) (unless noted otherwise)}$

20  $\frac{\frac{1}{g} \text{ Rate constants of k(OH + isoprene)} = 1.01 \times 10^{-10} \text{ cm}^3 \text{ molecule}^{-1} \text{ s}^{-1} \text{ O}_3 \text{ reaction rate constants determined in } 10^{-19} \text{ cm}^3}{\frac{1}{g} \text{ molecule}^{-1} \text{ s}^{-1} \text{ units. OH radical reaction rate constants determined in } 10^{-11} \text{ cm}^3 \text{ molecule}^{-1} \text{ s}^{-1} \text{ units. (Atkinson et al., 1990)}}{\frac{1}{g} \text{ nd} - \text{ no data.}}$

[revised manuscript text omitted]

a taken from Atkinson (Atkinson and Arey, 2003) unless noted otherwise.

b-Assumed OH radical concentration: 2.0x106 molecule cm-3, 12 h daytime average.

e-Assumed O3 concentration: 7x1044 molecule cm-3, 24 h average.

d Lifetimes are estimated in relation to [NO3] = 10 ppt, [O3] = 20 ppb for night; and [OH] = 106 molecules per cm3, [O3] = 20 ppb for day light conditions. (Kesselmeier and Staudt, 1999) (unless noted otherwise)

eRate constants (in units of  $10^{-17}$  cm3 molecule-1 s-1) for the gas- phase reactions of O3 with a monoterpenes have been determined at 296 ± 2 K and 740 torr total pressure of air or O2 using a combination of absolute and relative rate techniques. (Atkinson et al., 1990) (unless noted otherwise)

4 Rate constants (in units of 10-11 cm3 molecule-1 sec-1) for the gas phase reactions of the OH radical with monoterpenes have been determined in one atmosphere of air at 294 ± 1 K. (Atkinson et al., 1986) (unless noted otherwise)

Gas chromatography mass spectrometry (GC-MS) coupled with pre-concentration techniques has been developed to successfully identify and quantify different atmospheric monoterpenes (Janson, 1993; Räisänen et al., 2009; Song et al.,

 $\begin{cases} \frac{g}{Rate constants of k(OH + isoprene) = 1.01 \times 10^{-10} \text{ cm}^3 \text{ molecule}^{-1} \text{ s}^{-1} \text{ O}_3 \text{ reaction rate constants determined in } 10^{-19} \text{ cm}^3 \text{ molecule}^{-1} \text{ s}^{-1} \text{ units. OH radical reaction rate constants determined in } 10^{-14} \text{ cm}^3 \text{ molecule}^{-1} \text{ s}^{-1} \text{ units. OH radical reaction rate constants determined in } 10^{-10} \text{ cm}^3 \text{ molecule}^{-1} \text{ s}^{-1} \text{ units. OH radical reaction rate constants determined in } 10^{-10} \text{ cm}^3 \text{ molecule}^{-1} \text{ s}^{-1} \text{ units. (Atkinson et al., 1990)} \text{ nd - no data} \end{cases}$

2015). However, the requirements of pre-concentration and long cycle time (more than 1h) are obviously unsuitable for realtime measurements.

A promising approach to the near real time analysis of isomeric molecules is to combine both SCI-MS and fast GC methods. Pre-separation provided by fast GC requires involves short columns with thin active layers, fast temperature ramps, fast

- 5 injection systems and time resolutions below 5 min (Matisová and Dömötörová, 2003). Materic et al. (Materić et al., 2015) established a system using PTR-MS coupled with a fast GC to detect individual monoterpenes and achieved the separation of six most common monoterpenes at a limit of detection down to 1 ppbv. Pallozzi et al. then compared a fastCG-PTR-ToF-MS system with traditional GC-MS methods, discussing the limitations of the fast GC peak separationsetup on some BVOCs emitted from plants, including monoterpenes (Pallozzi et al., 2016). The authors then recommended applying longer columns
- 10 operating with fast temperature gradient such as 25 °C.min-1. SIFT-MS is also widely used in VOCs analyses (Allardyce et al., 2006; Smith and Španěl, 2011b, 2005b), which). It has well-defined analytical reaction conditions and the  $H_3O^+$ ,  $NO^+$  and  $O_2^{+\bullet}$  reagent ions can be switched rapidly to analyse time-varying trace gas concentrationsgases in air samples. In the present article, we report experimental developmentsmethod development results aimed at selectively analysingto selective analyses of individual monoterpenes in mixtures in air using a bespoke fast GC/SIFT-MS combination with  $H_3O^+$  and  $NO^+$
- 15 reagent ions. This involved the analysis of both prepared laboratory monoterpene/air mixtures and the headspace of the foliage of different pine trees.

Figure 1: Schematic visualization of the fast GC-SIFT-MS experiment. Coloured-dashed lines in the inlet part of the fast CG represent gas flow through the system of the valves EV1-3. The blue line traces the "normal mode" regime, the green line represents the "sampling mode" and the red line represents the "cleaning mode".

**2 Construction of a fast GC device for pre-separation**

- 5 The experimental setup of the bespoke fast GC setup constructed as an addition to SIFT-MS is shown in Fig. 1. The routing of the sample and the carrier gases was controlled by solenoid valves (Parker VSONC-2S25-VD-F, < 30ms response), labelled in Fig. 1 as EV1, EV2 and EV3. The needle valve NV1 was used in combination with an overflow relieve tube to fine-adjust the flow rate of the carrier gas (20-50 sccm from a gas cylinder, regulator set to about 2 bar) so that the air pressure at the column entrance is held just above ambient. The region of the sampling input line, EV2, EV3 and their</li>
- 10 connection with the column are permanently heated to ~60 °C to prevent adsorption of sample gas/vapour and to reduce memory effects.

Three modes of gas flow are possible as illustrated in Fig. 1:

- The "normal mode": EV2 is open and both EV1 and EV3 are closed. Carrier gas flows through NV1, partly vented via the overflow reliefrelieve but mostly into the column. The pressure at the column entrance is just above the ambient atmosphere and a constant flow rate of clean carrier gas (synthetic air or helium) is thus achieved.
- The "sampling mode": EV1 and EV2 are closed and EV3 is open. Sample air is introduced into the column in a short time (1 to 128 s) after which the "normal mode" is resumed.
- The "cleaning mode": All valves are open and the carrier gas taken directly from the cylinder regulator is introduced into the column (higher than normal flow) and purges the sample line via EV3. The overflow relieve
- 20

15

flow rate is not sufficient to diminish the pressure.

The modes can be switched either manually or controlled from the SIFT-MS software.

---

## Author Response (AR3)

Editor comments

*Dear Yoshi Iinuma,*

*Thank you for the list of specific comments. All of them have been acted upon.*

*Best regards,*

*Michal Lacko*

Page and line numbers are for the main text of author_response_version4.pdf

Page 2 Table 1 and elsewhere: Change "R-limonene" to "(R)- limonene".

Page 4 Figure 1, page 12 Table 2 footnote, page 17 line 5, page 23 line 18 and page 25 line 11:

Change "GC-SIFT-MS" to "GC/SIFT-MS".

Page 5 line 10: Change "°C.min-1" to "°C min-1".

Page 7 lines 12 and 14: Change "°um" to "μm".

Page 8 line 18: Change "Ω/m" to "Ω m-1".

Page 8 line 29: Change "cm.s-1" to "cm s-1".

Page 9 line 8: Change "a-pinene" to "α-pinene".

Page 9 lines 8 and 9: Change "KJ.mol-1" to "KJ mol-1".

Page 11 line 8: Change "L/mol" to "L mol-1".

Page 11 line 14: Change "cm3s-1" to "cm3s-1".

Page 11 line 14: Change "c/s" to "c s-1".

Page 11 line 18: Change "°um" to "μm".

Page 11 line 29: Change "ml" to "mL".

Page 17 lines 24 and 25: Remove a space between a number and a plus minus sign.

Page 17 line 31: Add "OriginLab Corporation, Northampton, MA, USA" for the citation of Origin 9.0.

Line 19 lines 17-32: Remove a space between a number and a plus minus sign.

[revised manuscript text omitted]